# A review of complementary principle of evaporation: From original linear relationship to generalized nonlinear functions

Songjun Han[1]*, Fuqiang Tian[2]

[1]State Key Laboratory of Simulation and Regulation of Water Cycle in River Basin, China Institute of Water Resources and Hydropower Research, Beijing 100038, China, hansj@iwhr.com
[2]State Key Laboratory of Hydro-science and Engineering, Tsinghua University, Beijing 100084, China,

*Correspondence to*: Songjun Han (hansj@iwhr.com)

**Abstract.** The complementary principle is an important methodology for estimating actual evaporation by using routinely observed meteorological variables. This review summaries its 56-year development, focusing on how related studies have shifted from adopting a symmetric linear complementary relationship (CR) to employing generalized nonlinear functions. The original CR denotes that the actual evaporation ($E$) and "apparent" potential evaporation ($E_{pa}$) depart from the potential evaporation ($E_{po}$) complementarily when the land surface dries from completely wet environment with constant available energy. The CR was then extended to an asymmetric linear relationship, and the linear nature was retained through properly formulating $E_{pa}$ and/or $E_{po}$. Recently, the linear CR was generalized to a sigmoid function and a polynomial function respectively. The sigmoid function does not involve the formulations of $E_{pa}$ and $E_{po}$, yet uses the Penman (1948)'s potential evaporation and its radiation component as inputs, whereas the polynomial function inherits $E_{po}$ and $E_{pa}$ as inputs and requires proper formulations for application. The generalized complementary principle has a more rigorous physical base and offers a great potential in advancing evaporation estimation. Future studies may cover several topics including the boundary conditions under wet environments, the parameterization and application over different regions of the world, and integrating with other approaches for further development.

## 1 Introduction

The complementary principle provides a framework for estimating terrestrial land surface evaporation by adopting routinely observed meteorological variables, and offers strong potential applications (Brutsaert and Stricker, 1979; McMahon et al., 2016; Morton, 1983). In this review paper, the terms "evaporation" and "evapotranspiration" are considered equivalent. As its underlying physical basis, this principle originated from the negative feedback of areal evaporation on evaporation demand (Bouchet, 1963; Brutsaert, 2015) as illustrated by the fact that reducing areal evaporation can make the overpassing air hotter and drier (Morton, 1983). By contrast, the Penman approach neglects above feedback (Morton, 1983), and relies heavily on land surface variables such as soil moisture content or stomatal resistance (Monteith, 1965; Penman, 1950). Compared to the Budyko framework (Budyko, 1974; Turc, 1954), which is often used for partitioning catchment evaporation from precipitation at mean annual or annual time scales, actual evaporation from the landscape can be estimated

within the complementary framework from nearly hourly to annual time scales. However, as a less popular hydro-climatic framework to estimate evaporation, the complimentary principle needs to be advanced not only for its own development, but also for the integration with other approaches for further development of evaporation research.

Based on the complementary principle, Bouchet (1963) first proposed a complementary relationship (CR) among three types of evaporation (Brutsaert, 2015), namely, the actual evaporation ($E$) from an extensive landscape under natural conditions, the apparent potential evaporation ($E_{pa}$) of a small saturated surface inside the landscape, and the potential evaporation ($E_{po}$) that occurs from the same large-size surface of $E$ when it is saturated. In practice, $E_{pa}$ corresponds to current atmosphere in contact with the unsaturated evaporating surface as the overpassing air is not affected by the small saturated surface, whereas the atmosphere corresponding to $E_{po}$ is in contact with the "potential" saturated surface. Thus, the surface water availability can be detected from the relative magnitude of $E_{pa}$ and $E_{po}$ because of the land surface-atmosphere interaction, and $E$ can be estimated without the explicit knowledges of the surface. The original symmetric linear "complementary" relationship (Bouchet, 1963; Brutsaert and Stricker, 1979; Morton, 1983) evolved into an asymmetric linear relationship (Brutsaert and Parlange, 1998; Pettijohn and Salvucci, 2006; Szilagyi, 2007). However, its development and applications are hindered by the use of complex formulations of $E_{po}$ and $E_{pa}$ to retain the linear CR, which will be reviewed in more detail in the following sections.

Recent studies have adopted the "generalized" complementary principle, which employs nonlinear functions instead of the linear CR (Brutsaert, 2015; Han et al., 2012; Han and Tian, 2018a). The generalized complementary function comes in two ways, with the first attempt abandons the concept of $E_{pa}$ and $E_{po}$ yet uses a sigmoid function to describe the relationship among $E$, Penman's potential evaporation ($E_{Pen}$), and its radiation term ($E_{rad}$) (Han et al., 2012; Han and Tian, 2018a). By contrast, the other attempt adopts a polynomial function to describe the relationship between $E$, $E_{pa}$ and $E_{po}$. However, $E_{pa}$ and $E_{po}$ still need to be formulated before applying the polynomial function to practical problems (Brutsaert, 2015).

The generalized complementary principle with earlier linear CRs as special cases has a more rigorous physical base (Brutsaert, 2015; Han and Tian, 2018b), and its methodology based on nonlinear functions is robust and effective. The generalized complementary principle has received much attention for its promising applications in estimating evaporation upon its proposal (Ai et al., 2017; Brutsaert et al., 2019; Brutsaert et al., 2017; Han and Tian, 2018a; Liu et al., 2016; Szilagyi et al., 2016; Zhang et al., 2017). However, the boundary conditions and proper mathematical forms of the generalized complementary functions are still under study (Crago et al., 2016; Han and Tian, 2018a; Ma and Zhang, 2017; Szilagyi et al., 2017). In this review, we summarize the 56-year development of the complementary principle with a specific focus on its evolution from a symmetric linear CR to generalized nonlinear functions. We also compare the two types of generalized complementary functions, and discuss their future development.

## 2 Linear complementary relationship

### 2.1 Concept of symmetric complementary relationship

The concept of CR is illustrated in Figure 1. When the water availability of the landscape is not limited, $E$ is
assumed to proceed at $E_{pa}$ and $E = E_{pa} = E_{po}$. Given that the surface dries with constant available energy, $E$ and $E_{pa}$
depart from $E_{po}$ with equal yet opposite changes in fluxes and exhibit a complementary relation as follows:

$$E_{pa} - E_{po} = E_{po} - E .$$ ( 1 )

The formulations of $E_{pa}$ and $E_{po}$ should be specified in Eq. (1). Bouchet (1963) assumed $E_{po}$ to be half the input
solar radiation. Morton (1976) calculated $E_{pa}$ by using the modified Penman's (1948) equation proposed by Kohler and
Parmele (1967)( $E_{Pen}^{KP}$ ), in which a constant vapor transfer coefficient was used to replace the wind function, and calculated
$E_{po}$ by using the Priestley–Taylor's (1972) equation ( $E_{PT}$ ) for an extensive saturated surfaced with minimal advection. This
method has been used to calculate monthly evaporation in large areas.

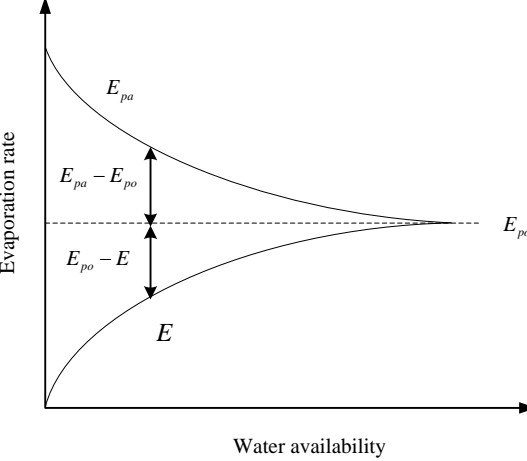

**Figure 1. Schematic illustration of symmetric CR following Boutchet (1963)**

**Table 1. Formulations of $E_{pa}$ and $E_{po}$ in different linear CR formulations**

| Types | $E_{pa}$ | $E_{po}$ | $b$ | References |
|---|---|---|---|---|
| | $E_{Pen}^{KP}$ | $E_{PT}$ | | Morton (1976) |
| | $E_{Pen}$ | $E_{PT}$ | | Brutsaert and Stricker (1979) |
| Symmetric | $E_{Mor}$ | $E_{PT}^{T_p}$ | 1 | Morton (1983) |
| | $E_{Pen}^{r_s}$ | $E_{Pen}^{r_s=0}$ | | McNaughton and Spriggs (1989) |
| | $E_{Pen}$ | $E_{PT} + \left| R_n - G - E_{Pen} \right|$ | | Parlange and Katul (1992a) |

| | $E_{pa}$ | $E_{po}$ | param | Reference |
|---|---|---|---|---|
| | $E_{PM}^{r_{s\min}}$ | $E_{PT}$ | | Pettijohn and Salvucci (2006) |
| | $E_{Pen}$ | $E_{PT}^{T_{ws}}$ | | Szilagyi and Jozsa (2008) |
| Asymmetric | $E_{Pan}$ | $E_{PT}$ | $b$ | Kahler and Brutsaert (2006) |
| | $ET_0$ | $E_{PT}$ | $b$ | Han et al. (2014d) |
| | $E_{MT}$ | $E_{Pen}$ | $\gamma/\Delta(T_a)$ | Granger (1989) |
| | $E_{Pen}$ | $E_{PT}$ | $\gamma/\Delta(T_s)$ | Szilagyi (2007) |
| | $E_{Pen}$ | $E_{PT}^{T_{ws}}$ | $f(RH)$ | Szilagyi (2015) |

*The symbols can be referred to the main text and the appendix for details.

Brutsaert and Stricker (1979) proposed the advection-aridity (AA) approach at a daily timescale, where $E_{pa}$ and $E_{po}$ are directly formulated by $E_{Pen}$ and $E_{PT}$, respectively. Although various combinations of $E_{pa}$ and $E_{po}$ exist (Table 1), $E_{po}$ is widely accepted to reflect the energy input while $E_{pa}$ includes the drying power of air simultaneously (Bouchet, 1963; Lhomme and Guilioni, 2006; Morton, 1983). Therefore, the AA approach seems logical and convincing (Lhomme and Guilioni, 2006). This approach has been validated based on hourly (Crago and Crowley, 2005; Parlange and Katul, 1992a), daily (Ali and Mawdsley, 1987; Brutsaert and Stricker, 1979; Qualls and Gultekin, 1997), monthly (Hobbins et al., 2001a; Lemeur and Zhang, 1990; Xu and Singh, 2005), and annual (Ramirez et al., 2005; Yu et al., 2009) data from either plot-scale lysimeters and eddy-covariance measurements or basin-wide water balance-derived results. By calculating $E_{Pen}$ and $E_{PT}$ using the standard meteorological data, the AA approach has been applied to estimate evaporation in regions with various land cover and climatic features (Hobbins et al., 2001a; Liu et al., 2006; Ozdogan and Salvucci, 2004; Wang et al., 2011). For instance, this approach has been applied and validated in China from the Gobi Desert with a mean annual precipitation of less than 150 mm (Han et al., 2008; Lemeur and Zhang, 1990; Liu and Kotoda, 1998) to the humid Eastern China with an annual precipitation of approximately 1,800 mm (Xu and Singh, 2005). Note that however, the AA approach tends to overestimate $E$ under wet environments but underestimate $E$ under arid environments. Measurement error, imperfect formulations of $E_{pa}$ and/or $E_{po}$, external energy sources, or even the nonlinear nature of the complementary principle were considered as potential causes of this bias (Han et al., 2008, 2012; Hobbins et al., 2001a; Qualls and Gultekin, 1997).

**2.2 Proofs of complementary relationship**

Bouchet (1963) and Morton (1970; 1965) approximately validated the CR by using annual and monthly data, respectively. At an annual scale, $E$ and $E_{pa}$ (which are represented by $E_{Pen}$ or pan evaporation ($E_{Pan}$)) were plotted against annual precipitation and their negative relationship was used as an evidence to support the reliable probability of the complementarity (Morton, 1983). Ramirez et al. (2005) tested the CR by using a composite of 192 data pairs from 25 basins across US, and claimed a direct observational evidence. Yu et al. (2009) examined the CR at 102 observatories across China and found the CR at low elevations. Su et al. (2015) also showed a negative correlation between $E$ from atmospheric

reanalysis data and $E_{Pan}$ in the non-humid regions of China. The large scale irrigation development in an arid environment provides a large "natural" experimental area for validating the CR by the opposite changes in $E$ and $E_{pa}$ (Roderick et al., 2009). A study from Turkey revealed that the warm-season $E_{pa}$ decreased progressively along with an increasing irrigated area (Ozdogan and Salvucci, 2004). Similar results were obtained from arid irrigation districts in Northwest China, where an increasing irrigation water consumption reduces $E_{pa}$ (Han et al., 2014d) whereas a decreasing irrigation water consumption increases $E_{pa}$ (Han et al., 2017). However, although these studies showed that $E$ and $E_{pa}$ move in opposite directions in most cases, there was not solid evidence to support the symmetric nature of CR.

The plausibility of CR has also been validated on theoretical bases and has been mathematically rationalized by Bouchet (1963), Morton (1969); Morton (1971), and Seguin (1975). The rationalization proposed by Morton (1969, 1971) considers the governing changes in the humidity and temperature of the equilibrium sublayer of the atmospheric boundary layer (ABL) by assuming that (1) the net radiation will not change with the surface, and (2) the heat and vapor eddy transfer characteristics are identical for $E$ and $E_{pa}$. Relaxing the second assumption of Morton (1983), Szilagyi (2001) derived the CR by using the mass conservation equation for water vapor. However, LeDrew (1979) argued that Morton's two assumption do not necessarily hold, and pointed out that the symmetric CR is physically unrealistic by using a diagnostic model of the energy fluxes within a closed system.

The physical basis of the CR has been further explored by using climate models. McNaughton and Spriggs (1989) tested the CR by using a simple model of the atmospheric mixed layer with entrainment in which the latent heat of the surface is simulated by using the bulk mass transfer equation with bulk resistance. During the validation, $E_{pa}$ is calculated via Penman's equation, which uses the temperature and humidity obtained from the results of the mixed-layer model corresponding to certain resistance ($E_{Pen}^{r_s}$), while $E_{po}$ is calculated with the surface resistance set to zero ($E_{Pen}^{r_s=0}$). Kim and Entekhabi (1997) added the surface energy balance and atmospheric thermal radiation fluxes into the model to extend the study of McNaughton and Spriggs (1989). By using the Penman–Monteith equation to govern the areal latent heat flux at the surface, Lhomme (1997b) proposed a closed-box model with an impermeable lid at a fixed height while Lhomme (1997a) used a more realistic open-box model of the ABL with entrainment to assess the CR. Sugita et al. (2001) tested the CR by using a modified version of Lhomme (1997a)'s model, which was calibrated by using a dataset obtained from the Hexi Corridor desert area in Northwest China. But a strict symmetric CR was hardly confirmed by these studies.

## 2.3 Asymmetric linear complementary relationship

With $E_{pa}$ and $E_{po}$ denoted by the mass-transfer type potential evaporation $E_{MT}$ and $E_{Pen}$, respectively, Granger (1989) proposed an alternative CR as follows:

$$(E_{MT} - E_{Pen}) = \frac{\Delta(T_a)}{\gamma}(E_{Pen} - E), \tag{2}$$

where $\gamma$ is the psychrometric constant, $\Delta(T_a)$ is the slope of the saturation vapor pressure at air temperature $T_a$. Despite being identical to the surface energy balance, Eq. (2) has inspired researchers to examine whether the CR should be symmetric or not (Pettijohn and Salvucci, 2006; Szilagyi, 2007). By using pan evaporation to denote $E_{pa}$, Brutsaert and Parlange (1998) extended the symmetric CR as follows:

$$(E_{pa} - E_{po}) = b(E_{po} - E), \tag{3}$$

where $b$ is the coefficient that denotes asymmetry, and the original symmetric CR is characterized with $b$=1. Kahler and Brutsaert (2006) clarified and tested Eq. (3) at a daily timescale and attributed the asymmetry to the nature of the heat transfer between the pan and its surroundings, which made the changes in pan evaporation larger than those in $E$. Szilagyi (2007) showed that the asymmetry is not limited only to the evaporation pan but is inherently linked to the definition of $E_{pa}$. Brutsaert (2015) stated that asymmetry is an inherent characteristic of the CR.

The asymmetric CR can be illustrated in a dimensionless form (Figure 2) (Kahler and Brutsaert, 2006). Normalized by $E_{po}$, $E_{pa}$ and $E$ can be scaled as

$$\frac{E}{E_{po}} = \frac{(1+b)\,E/E_{pa}}{1+b\,E/E_{pa}} \quad \text{and} \quad \frac{E_{pa}}{E_{po}} = \frac{1+b}{1+b\,E/E_{pa}}. \tag{4}$$

The scaled $E_{pa}$ and $E$ are both functions of the dimensionless variable $E/E_{pa}$, while $E/E_{pa}$ serves as the evaporative surface moisture index. Compared with the original form (Eq. (1) and Figure 1), the CR here is illustrated without the presence of the water availability explicitly. The asymmetric CR has been validated via the opposite changes of $E/E_{po}$ and $E_{pa}/E_{po}$ against $E/E_{pa}$ at several locations over the world. However, the wet conditions were seldom explored, which may hide the true correlation as the two curves of $E/E_{po}$ and $E_{pa}/E_{po}$ approach.

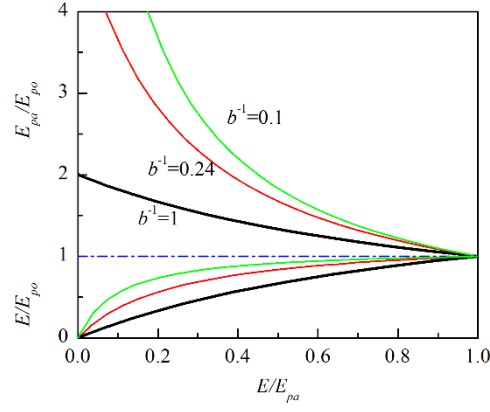

**Figure 2. Scaled $E_{pa}$ and $E$, which serve as functions of the evaporative moisture index $E/E_{pa}$ and calculated on the basis of the asymmetric CR, according to method of Kahler and Brutsaert (2006).**

The asymmetric CR is a significant improvement of the symmetric CR, and the opposite changes of $E/E_{po}$ and $E_{pa}/E_{po}$ against $E/E_{pa}$ were treated as an enhanced illustration of the CR (Brutsaert et al., 2019; Hu et al., 2018; Ma et al., 2015a; Szilagyi, 2007; Zhang et al., 2017). The performances on evaporation estimation are improved by calibrating the asymmetry parameter *b* (Han et al., 2008; Huntington et al., 2011; Kahler and Brutsaert, 2006; Ma et al., 2015a). Efforts have also made to calculate *b* by using the meteorological variables, which enhance the predict ability of the CR (Aminzadeh et al., 2016; Szilagyi, 2015; Szilagyi, 2007). However, the changes in *b* imply a potential nonlinear characteristic of the CR (Han, 2008; Lintner et al., 2015). The observed values of $E/E_{po}$ and $E_{pa}/E_{po}$ can even exhibit a positive correlation under wet conditions at several flux sites, which challenges the linear CR (Han and Tian, 2018a).

### 2.4 Efforts in retaining the linear nature of complementary relationship through properly formulating $E_{pa}$ and/or $E_{po}$

The imperfect linear CR has inspired researchers to apply rational formulations of $E_{pa}$ and/or $E_{po}$ to retain it. One direct method is to revise the formulations of $E_{Pen}$ and/or $E_{PT}$ based on the AA approach through calibration. For $E_{Pen}$, the empirical wind function was calibrated to improve the CR (Hobbins, 2001). However, Penman's wind function cannot work under the wet and dry conditions simultaneously (Pettijohn and Salvucci, 2006). The wind function derived from Monin–Obukhov's similarity theory was then employed (Crago and Crowley, 2005; Ma et al., 2015a; Parlange and Katul, 1992b; Pettijohn and Salvucci, 2006). The surface roughness and surface albedo were also calibrated to improve the CR (Lemeur and Zhang, 1990). Meanwhile, for $E_{PT}$, the Priestley–Taylor coefficient ($\alpha$) is regarded varying, thereby leaving a range for calibration (Han et al., 2006; Xu and Singh, 2005; Yang et al., 2012). In addition to $E_{Pen}$ and $E_{PT}$, the mass-transfer type potential evaporation (van Bavel, 1966) ($E_{MT}$) was considered as another formulation of $E_{pa}$ (Granger, 1989). Different combinations of $E_{pa}$ and $E_{po}$, (i.e., $E_{Pen}$, $E_{PT}$, and $E_{MT}$) were tested through the trial-and-error method to retain the linear nature of CR (Anayah and Kaluarachchi, 2014; Crago and Crowley, 2005).

Given the conceptual inadequacy in using $E_{Pen}$ and $E_{PT}$ to denote $E_{pa}$ and $E_{po}$ (Morton, 1983; Szilagyi and Jozsa, 2008), a better CR must be obtained by modifying the formulations of $E_{Pen}$ and/or $E_{PT}$ on physical basis. For $E_{pa}$, the net long-wave radiation depends on the land surface temperature; meanwhile, adjusting surface temperature with air temperature to calculate solar radiation in $E_{Pen}$ may be problematic (Morton, 1983). To address these limitations, Morton (1983) combined the energy balance and water vapor transfer equations by using an equilibrium temperature ($T_p$) and derived a Morton-type potential evaporation $E_{Mor}$ to denote $E_{pa}$. By attributing the asymmetry to the assumption that $E_{pa}$ conceptually includes a transpiration component, Pettijohn and Salvucci (2006) improved the asymmetry by replacing $E_{Pen}$

with the Penman–Monteith equation with a minimum surface resistance ( $E_{PM}^{r_{s\,\min}}$ ). Similarly, the reference evapotranspiration ( $ET_0$ ) was also used to replace $E_{Pen}$ (Han et al., 2014d; Han et al., 2017).

In theory, $E_{po}$ is the potential evaporation when the land surface is saturated, and should be calculated with a proper formula by using meteorological variables corresponding to the "potential" saturated surface. The Priestley-Taylor equation has been widely accepted to represent evaporation from extensive saturated surfaces by using meteorological variables corresponding to these saturated surfaces (Brutsaert, 1982; Priestley and Taylor, 1972). This way it was suggested to represent $E_{po}$ (Brutsaert and Stricker, 1979). However, in the AA approach, $E_{PT}$ is calculated by the Priestley-Taylor equation using the atmospheric variables that correspond to the current unsaturated surface. But the atmosphere in contact with the land surface will change if the land surface was brought into saturated (Brutsaert, 2015; Morton, 1983). Thus, $E_{PT}$ is in reality a variable dependent on the meteorological variables at the time of calculation and does not represent the "true" $E_{po}$.

Obviously, calculating the slope of the saturation vapor pressure at the current air temperature ( $\Delta(T_a)$ ) for $E_{PT}$ is imperfect because the temperature corresponding to $E_{po}$ is different from current $T_a$ corresponding to an unsaturated environment (Morton, 1983; Szilagyi and Jozsa, 2008). Thus, predicting the air temperature corresponding to the extensive saturated surface is critical for properly formulating $E_{po}$. Morton (1983) derived $E_{po}$ by using a modified Priestley–Taylor equation with net radiation and the slope of the saturation vapor pressure that is calculated at equilibrium temperature $T_p$ ( $E_{PT}^{T_p}$ ). Szilagyi and Jozsa (2008) argued that $\Delta$ in $E_{PT}$ should be calculated at the air temperature corresponding to the wet environment instead of actual air temperature, which is not straightforward to derive. Thus, Szilagyi and Jozsa (2008) proposed an iterative approach based on the Bowen ratio method to estimate the surface temperature under wet environments ( $T_{ws}$ ), and replaced $\Delta(T_a)$ with the slope of the saturation vapor pressure curve at $T_{ws}$ ( $\Delta(T_{ws})$ ) in the Priestley–Taylor equation ( $E_{PT}^{T_{ws}}$ ) by assuming a negligible temperature gradient over such a small wet area. $E_{PT}^{T_{ws}}$ was used to improve the symmetry of the CR in arid shrubland environments (Huntington et al., 2011) and in an alpine steppe of the Tibetan Plateau (Ma et al., 2015a). The evaporation estimations across the US were also improved by applying the modified AA approach (Szilagyi, 2015; Szilagyi et al., 2009; Szilagyi and Jozsa, 2008). Aminzadeh et al. (2016) derived a steady state surface temperature via the surface energy balance at which the sensible heat flux is zero, and calculated $E_{pa}$ and $E_{po}$ using a mass-transfer type reference evaporation corresponding to current and saturated surface water content.

Advection is another factor influencing $E_{po}$, which could not be adequately considered by $E_{PT}$ with an assumption of a minimal advection effect (Morton, 1975, 1983; Parlange and Katul, 1992a). The effects of advection were considered by an empirical correction factor in $E_{PT}^{T_p}$ (Morton, 1975, 1983). Parlange and Katul (1992a) attributed the asymmetry to the

horizontal advection of dry air, which would make $E_{Pen}$ larger than the available energy ($R_n - G$) (i.e., $R_n - G - E_{Pen} < 0$) and proposed to replace $E_{PT}$ with $E_{PT} + |R_n - G - E_{Pen}|$ to improve the CR on an hourly basis.

The efforts of reformulating $E_{pa}$ and/or $E_{po}$ through the calibration, trial-and-error process and the physical improvement have significantly improved the evaporation estimation (Hobbins et al., 2001b; Ma et al., 2015a; Szilagyi, 2015; Xu and Singh, 2005). However, it is always impossible to find formulations of $E_{pa}$ and $E_{po}$ completely rational at present, and these approaches are deemed ineffective because of their high computation demand, which is a key stumbling block when applying the CR at large-scale (e.g., continental or global) (Ma et al., 2019).

## 3 Generalized complementary principle via nonlinear functions

### 3.1 Normalized complementary functions

Unlike the normalization by $E_{po}$ (Kahler and Brutsaert, 2006), Han (2008) normalized Eq. (3) by using $E_{pa}$ and found that $E/E_{pa}$ is expressed as a linear function of $E_{po}/E_{pa}$. Normalized by $E_{Pen}$ (Han et al., 2008), the AA approach can be expressed as

$$\frac{E}{E_{Pen}} = \alpha(1 + \frac{1}{b})\frac{E_{rad}}{E_{Pen}} - \frac{1}{b}, \tag{5}$$

where $E/E_{Pen}$ is regarded as a linear function of $E_{rad}/E_{Pen}$. The bias of the AA function under arid and wet environments can be easily understood in its dimensionless form. Also, the AA approach with a tuned $b$ still underestimated evaporation in arid environments (Han et al., 2008), which implies that the CR may deviate from its linear characteristics.

Based on the examination of the CR using a model of the convective boundary-layer with entrainment (Lhomme, 1997a), Lhomme and Guilioni (2006, 2010) recommended a form of the CR through the effective surface resistance of the region. Integrating this relationship into Penman–Monteith equation and the normalization by $E_{Pen}$ lead to

$$\frac{E}{E_{Pen}} = (1 + \omega)\frac{E_{rad}}{E_{Pen}}, \tag{6}$$

where $\omega$ is a coefficient accounting for the entrainment of dry air within the atmospheric boundary layer. Equation (6) is a linear function without intercept, but was not verified and applied using observed data.

The CR model proposed by Granger (1989) based on Eq. (2) has demonstrated promising application across different land covers and regional climate conditions (Carey et al., 2005; Granger, 1999; Granger and Gray, 1989b; Pomeroy et al., 1997; Xu and Singh, 2005). In fact, the relationship between relative evaporation and relative drying power plays a key role in reflecting the dryness of the surface (Granger and Gray, 1989a). This relationship was integrated to a asymmetric CR to improve the performance on evaporation estimation (Anayah and Kaluarachchi, 2014). Normalized by $E_{Pen}$, Granger's

model is similar to the AA function in that $E/E_{Pen}$ is expressed as a function of the relative magnitude of drying power to net radiation (Han et al., 2011). By synthesising the dimensionless forms of the AA function and the Granger's model, Han et al. (2011) proposed the following function as an alternative:

$$\frac{E}{E_{Pen}} = \frac{1}{1 + c_1 e^{d(1 - \frac{E_{rad}}{E_{Pen}})}}, \tag{7}$$

where $c_1$ and $d$ are the parameters. Eq. (7) approximates the linear AA function under normal conditions neither too wet nor too dry but amends its bias (Han et al., 2011), thus can be regarded an enhanced nonlinear version of the linear CR.

Actual evaporation can be estimated using routinely measured meteorological data by using the climatological resistance to parameterize the bulk surface resistance in the Penman–Monteith equation (Katerji and Perrier, 1983; Liu et al., 2012; Ma et al., 2015b; Rana et al., 1997). A linear relationship between the ratio of surface resistance to aerodynamic resistance and the ratio of climatological resistance to aerodynamic resistance was proposed by Katerji and Perrier (1983). Han et al. (2014c) integrated this linear relationship into the Penman–Monteith equation and derived a dimensionless form via normalization by $E_{Pen}$:

$$\frac{E}{E_{Pen}} = \frac{1}{1 + k(\frac{E_{Pen}}{E_{rad}} - 1) + l}, \tag{8}$$

where $k$ and $l$ are the empirical calibration parameters. With similar variables yet different mathematical formulations, Eq. (8) can also be considered a complementary function (Han et al., 2014c).

## 3.2 Sigmoid function relating $E/E_{Pen}$ to $E_{rad}/E_{Pen}$

By synthesizing the aforementioned functions (Table 2), Han et al. (2012) generalized the CR as a function that relates $E/E_{Pen}$ to $E_{rad}/E_{Pen}$:

$$E/E_{Pen} = f(E_{rad}/E_{Pen}). \tag{9}$$

Eq. (9) shares the same form of Penman's approach for estimating evaporation. The function of surface wetness that denotes the reduction of $E$ to $E_{Pen}$ is replaced by the function of $E_{rad}/E_{Pen}$, which is termed "atmospheric wetness" (Han and Tian, 2018b). Despite not explicitly exhibiting a CR, Eq. (9) holds the complementary principle that the land surface wetness is indirectly denoted by the drying power of air with a constant radiation energy input (Brutsaert, 1982). Accordingly, Eq. (9) is considered a "general form" of the CR (Han et al., 2014b) (hereinafter referred to as H12 whereas the other type of generalized complementary function first proposed by Brutsaert (2015) if referred to as B15 for comparison). The existing analytical forms of the function can be classified into linear, concave/convex, or sigmoid (Table 2). Studies on the complementary principle can be advanced by formulating a proper analytical form for H12.

**Table 2. Different analytical formulas for generalized complementary function H12**

| Type | Formula[*] | References |
|------|-----------|------------|
| Linear | $y = \alpha(1+\dfrac{1}{b})x - \dfrac{1}{b}$ | Brutsaert and Stricker (1979) |
| | $y = (1+\omega)x$ | Lhomme and Guilioni (2006, 2010) |
| Sigmoid | $y = \dfrac{1}{1 + c_1 e^{d(1-x)}}$ | Granger (1989), Han et al. (2011) |
| | $y = \dfrac{1}{1 + m(\dfrac{1}{x}-1)^n}$ | Han et al. (2012) |
| | $y = \dfrac{1}{1 + m(\dfrac{x_{max}-x}{x-x_{min}})^n}$ | Han and Tian (2018a) |
| Concave/ convex | $y = \dfrac{1}{1 + k(\dfrac{1}{x}-1)+l}$ | Katerji and Perrier (1983) , Han et al. (2014b) |
| | $y = (2-c)\alpha^2 x^2 - (1-2c)\alpha^3 x^3 - c\alpha^4 x^4$ | Brutsaert (2015) |

[*] $x = E_{rad}/E_{Pen}$ and $y = E/E_{Pen}$ . the other symbols are parameters.

The exact analytical form of H12 is inadequately understood at present. However, some of its characteristics can be detected from its boundary conditions under extremely arid and completely wet environments. Han et al. (2012) derived the
zero-order and first-order boundary conditions for H12 as

$$
\begin{cases}
y_H = 0, \; x_H \to 0 \\
y_H = 1, \; x_H \to 1 \\
\dfrac{dy_H}{dx_H} = 0, \; x_H \to 0 \; , \\
\dfrac{dy_H}{dx_H} = 0, \; x_H \to 1
\end{cases}
\tag{10}
$$

where $x_H = E_{rad}/E_{Pen}$ and $y_H = E/E_{Pen}$ . Han et al. (2012) proposed the following sigmoid function (hereinafter this specific analytical form of H12 is referred to as sigmoid H2012):

$$
\frac{E}{E_{Pen}} = \frac{1}{1 + m(\dfrac{E_{Pen}}{E_{rad}}-1)^n} \; ,
\tag{11}
$$

where $m$ and $n$ are parameters. The results obtained from an extremely dry desert and a wet farmland reveal that the sigmoid H2012 corrects the bias of the linear AA and Equation (7) (Han et al., 2012); the application of this sigmoid function has also been recommended for an alpine meadow region of the Tibetan Plateau (Ma et al., 2015b).

**Table 3. Different forms of the generalized complementary function, $y = f(x)$**

| Approach | Specific function | $E_{po}$ | $x$ | $y$ | Typical type | Reference |
|---|---|---|---|---|---|---|
| H12[*] | H2017 | Not involved | $\dfrac{E_{rad}}{E_{Pen}}$ | $\dfrac{E}{E_{Pen}}$ | Sigmoid | Han et al. (2018) |
| | B2015 | $E_{PT}$ | $\dfrac{\alpha E_{rad}}{E_{Pen}}$ | $\dfrac{E}{E_{Pen}}$ | 4-order polynomial | Brutsaert (2015) |
| B15[**] | C2016 | $E_{PT}^{T_{ws}}$ | $\dfrac{E_{PT}^{T_{ws}}/E_{Pen} - E_{PT}^{T_{ws}}/E_{MT}^{max}}{1 - E_{PT}^{T_{ws}}/E_{MT}^{max}}$ | $\dfrac{E}{E_{Pen}}$ | Linear | Crago et al. (2016) |
| | S2017 | $E_{PT}^{T_{ws}}$ | $\dfrac{E_{Pen}^{max} - E_{Pen}}{E_{Pen}^{max} - E_{PT}^{T_{ws}}} \dfrac{E_{PT}^{T_{ws}}}{E_{Pen}}$ | $\dfrac{E}{E_{Pen}}$ | 3-order Polynomial | Szilagyi et al. (2017) |

[*]H12 denotes the t generalized complementary function $E/E_{Pen} = f(E_{rad}/E_{Pen})$, while H2012 and H2017 are the sigmoid
analytical formulas for H12.
[**]B15 denotes the generalized complementary function $E/E_{pa} = f(E_{po}/E_{pa})$, while B2015, C2016 and S2017 are the analytical formulas for B15 in application.

The zero-order arid boundary condition of H12 adopted in H2012 may be imperfect in the sense that the aerodynamic term ($E_{aero}$) of $E_{Pen}$ may not reach infinity under an arbitrary $E_{rad}$ (Crago et al., 2016; Kovács, 1987; Szilagyi et al., 2017).
Moreover, $E_{rad}/E_{Pen}$ cannot easily approach unity because of advection (Kovács, 1987; Priestley and Taylor, 1972). Therefore, Han and Tian (2018a) brought in the minimum and maximum limits of $E_{rad}/E_{Pen}$ ($x_{min}$ and $x_{max}$) under an assumed constant $E_{rad}$ and re-derived the boundary conditions of H12 by adopting two widely accepted assumptions following Penman's combination theory, namely, $\partial E/\partial E_{Pen} = 0$ under extremely arid environments and $E = E_{Pen}$ under completely wet environments. The boundary conditions are set as follows:

$$\begin{cases} y_H = 0, \ x_H \to x_{min} \\ y_H = 1, \ x_H \to x_{max} \\ \dfrac{dy_H}{dx_H} = 0, \ x_H \to x_{min} \\ \dfrac{dy_H}{dx_H} = 0, \ x_H \to x_{max} \end{cases}. \tag{12}$$

Based on the boundary conditions, Han and Tian (2018a) speculated that the growth of $E/E_{Pen}$ upon $E_{rad}/E_{Pen}$ exhibits a sigmoid feature, which is a three-stage pattern in which $E/E_{Pen}$ gradually increases along with $E_{rad}/E_{Pen}$, rapidly increases along with $E_{rad}/E_{Pen}$ in the following stage, and then demonstrates a decelerated growth in the final stage. The sigmoid feature can be detected from the study by Han et al. (2012) in the arid Gobi-HEIFE site and the humid Wudaogou site in China. Han and Tian (2018a) further validated the sigmoid feature according to the much larger regression slopes of $E/E_{Pen}$ upon $E_{rad}/E_{Pen}$ in the middle stage than those in the other two stages with smaller or larger values of $E_{rad}/E_{Pen}$ by using 22 eddy covariance towers from the FLUXNET (Baldocchi et al., 2001) dataset which includes representative biomes of grasslands, croplands, shrublands, evergreen needleleaf forests, deciduous broadleaf forests, and wetlands.

In 2017, Han and Tian (2018a) proposed the following new sigmoid function to accordance with the boundary conditions (hereinafter referred to as sigmoid H2017):

$$\frac{E}{E_{Pen}} = \frac{1}{1 + m(\dfrac{x_{\max} - E_{rad}/E_{Pen}}{E_{rad}/E_{Pen} - x_{\min}})^n} , \tag{13}$$

where $E_{rad}/E_{Pen}$ adopts the feasible domain ($x_{\min}$, $x_{\max}$), which is a subdomain of (0, 1). Both the linear AA function and sigmoid H2012 are special cases of sigmoid H2017. Han and Tian (2018a) performed a first-order Taylor expansion of Eq. (13) at the point where $E/E_{Pen} = 0.5$ and set the linear equation equivalent to the linear AA function. Afterward, the parameters $m$ and $n$ of sigmoid H2017 can be transformed from the Priestley–Taylor coefficient $\alpha$ and parameter $b$ of the AA function.

Han et al. (2008) was the first to plot the AA function as a linear in the state space ($E_{rad}/E_{Pen}$, $E/E_{Pen}$), in which the biases of the AA function under arid and wet environments can be understood easily. The analytical forms of the generalized complementary function H12 listed in Table 2 can be plotted as curves in a 2D space ($E_{rad}/E_{Pen}$, $E/E_{Pen}$) (Fig. 3), and the upper limits of $E_{Pen}$ and $E_{PT}$ are illustrated as the curve of *OMN*. The sigmoid H2012 was compared to the linear AA in the state space ($E_{rad}/E_{Pen}$, $E/E_{Pen}$) to demonstrate its improvement (Han et al., 2012). Observed $E/E_{Pen}$ can be plotted against $E_{rad}/E_{Pen}$ and fitted by the analytical functions of H12 in the state space ($E_{rad}/E_{Pen}$, $E/E_{Pen}$), which is an obvious improvement compared to the schematic illustrations of the CR in Fig. 1 and 2.

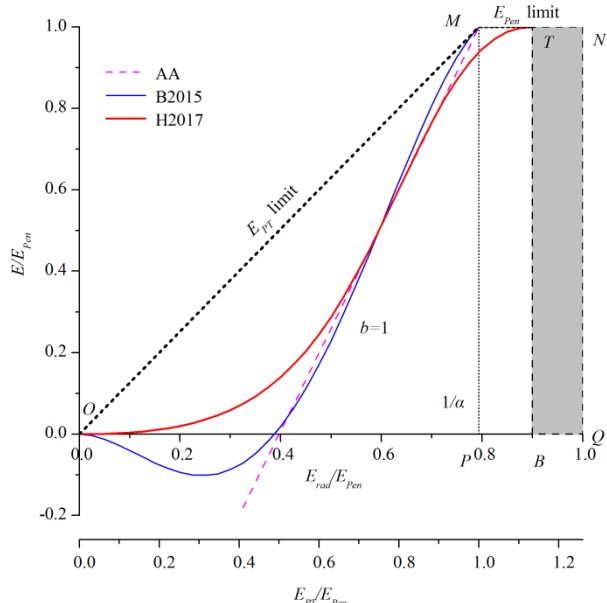

Figure 3. Generalized complementary functions in the state space ($E_{rad}/E_{Pen}$, $E/E_{Pen}$): linear AA, polynomial B2015, and sigmoid H2017, with $\alpha = 1.26$ and b=1. $x_{min}$ and $x_{max}$ are set to 0 and 0.9 respectively, as revised from Han and Tian (2018). *OM* is the edge at which $E = E_{PT}$, , *MN* is the edge where $E = E_{Pen}$, *M* corresponds to the condition of the minimal advection evaporation where $E_{PT} = E_{Pen}$ and *N* corresponds to the condition of the equilibrium evaporation where $E_{Pen} = E_{rad}$.

### 3.3 Polynomial function relating $E/E_{pa}$ to $E_{po}/E_{pa}$

Inspired by Han *et al.* (2012), Brutsaert (2015) reformulated another general dimensionless form of the CR, $E/E_{pa} = f(E_{po}/E_{pa})$, and proposed its boundary conditions as follows:

$$\begin{cases} y_B = 0, \ x_B \to 0 \\ y_B = 1, \ x_B \to 1 \\ \dfrac{dy_B}{dx_B} = 0, \ x_B \to 0 , \\ \dfrac{dy_B}{dx_B} = 1, \ x_B \to 1 \end{cases} \tag{14}$$

where $x_B = E_{po}/E_{pa}$ and $y_B = E/E_{pa}$. The following fourth-order polynomial function was also derived to satisfy the boundary conditions:

$$\frac{E}{E_{pa}} = (2-c)\left(\frac{E_{po}}{E_{pa}}\right)^2 - (1-2c)\left(\frac{E_{po}}{E_{pa}}\right)^3 - c\left(\frac{E_{po}}{E_{pa}}\right)^4 , \tag{15}$$

where $c$ is a parameter. Brutsaert (2015) regarded Eq. (15) (hereinafter referred to as B15) as a generalization of the linear CR and referred to the corresponding methodology as the "generalized complementary principle."

The application of Eq. (15) depends on specific formulations of $E_{pa}$ and $E_{po}$. In the manner of the AA approach, Eq. (15) has been applied to estimate evaporation (Ai et al., 2017; Brutsaert et al., 2017; Liu et al., 2016; Szilagyi et al., 2016; Zhang et al., 2017). In this case, we refer to Eq. (15) in the manner of the AA approach as B2015 to avoid confusion. Although estimating $E_{pa}$ by using $E_{Pen}$ is widely accepted by the research community, prognostically predicting $E_{po}$ based on $E_{PT}$ remains a huge challenge considering the theoretical problems of the Priestley-Taylor coefficient. In addition, the lower limit of $x_B \to 0$ of B15 may not hold in the manner of the AA approach (Crago et al., 2016; Han and Tian, 2018a; Kovács, 1987; Szilagyi et al., 2017). To address these challenges, Crago et al. (2016); Szilagyi et al. (2017) used the maximum value of $E_{pa}$ to rescale $x_B$ and replaced $E_{PT}$ with $E_{PT}^{T_{ws}}$, the latter of which is based on the air temperature in a wet environment. Crago et al. (2016) applied a mass transfer approach to calculate the maximum value of $E_{pa}$ ( $E_{MT}^{max}$ ) and rescaled $x_B$ as

$$x_C = \frac{E_{PT}^{T_{ws}} / E_{Pen} - E_{PT}^{T_{ws}} / E_{MT}^{max}}{1 - E_{PT}^{T_{ws}} / E_{MT}^{max}} \ . \tag{16}$$

Szilagyi et al. (2017) employed the Penman equation to calculate the maximum value of $E_{pa}$ ( $E_{Pen}^{max}$ ) and proposed the following rescaled version:

$$x_S = \frac{E_{Pen}^{max} - E_{Pen}}{E_{Pen}^{max} - E_{PT}^{T_{ws}}} \frac{E_{PT}^{T_{ws}}}{E_{Pen}} \ . \tag{17}$$

$x_C$ and $x_S$ are essentially same (Szilagyi et al., 2017) except for the different formulations for the maximum value of $E_{pa}$. However, $E_{MT}^{max}$ in Eq. (16) may became invalid under conditions with relatively strong available energy yet weak winds (Ma and Zhang, 2017), and was replaced with $E_{Pen}^{max}$ (Crago and Qualls, 2018) in the latest version. After rescaling, Crago et al. (2016) proposed a new linear version of the generalized complementary function (hereinafter referred to as C2016) (i.e., $y_B = x_C$; Table 2), while Szilagyi et al. (2017) used the third order polynomial function (hereinafter referred to as S2017) by replacing B15 with c=0. With the same independent variable yet different functions (Table 2), C2016 and S2017 demonstrate improvements in their evaporation estimation performance (Crago and Qualls, 2018; Crago et al., 2016; Szilagyi et al., 2017).

### 3.4 Comparisons between the two generalized complementary approaches

The two generalized complementary approaches, H12 and B15, are essentially different, with completely different normalized variables (Table 3). The differences in the analytical forms, sigmoid and 4-order polynomial, mainly result from

their wet boundary conditions. B15 inherits the concept of the three types of evaporation dated from the original CR, and its boundary conditions and analytical form are derived for $x_B = E_{po}/E_{pa}$ and $y_B = E/E_{pa}$. The original CR adopts the limits of $E_{pa}$ and $E_{po}$ on $E$ in a serial manner ($E \leq E_{po} \leq E_{pa}$) (Brutsaert, 2015) while considering that the wet regional evaporation must always be smaller than the wet patch evaporation ($E_{po} \leq E_{pa}$). Under wet conditions, B15 adopts $dy_B/dx_B = 1$ as $x_B \rightarrow 1$ by considering that any change in $E$ is the same as the change in $E_{po}$, which results in a concave polynominal type function. The limits and boundary conditions of B15 would be appropriate in theory. However, $E_{po}$ and

$E_{pa}$ should be formulated before B15 is applied to practical problems. Thus, B15 still faces one of the difficulties that the original CR has, that is, appropriately formulating $E_{po}$ and $E_{pa}$, which determines the validity and application of B15. So, future studies can be conducted towards more proper formulations of $E_{pa}$ and $E_{po}$ to satisfy the boundary conditions of B15.

By contrast, H12 goes further from the original CR. The boundary conditions and the analytical form of H12 are derived for $x_H = E_{rad}/E_{Pen}$ and $y_H = E/E_{Pen}$. The knowledge on $E_{po}$ is unnecessary, and only the mostly accepted $E_{Pen}$

and its radiation term appear in H12. By doing so, the corresponding theoretical and practical difficulties of formulating $E_{po}$ and $E_{pa}$ are eliminated. H12 adopts $E_{Pen}$ as the upper limit $E \leq E_{Pen}$ during the derivation and introduce the limit of $E \leq E_{PT}$ by considering that $E_{PT}$ is widely used as an upper limit of $E$ in practice. Han and Tian (2018a) showed that the upper limits of $E_{Pen}$ and $E_{PT}$ on evaporation must be in parallel, that is, $\begin{cases} E \leq E_{Pen} \\ E \leq E_{PT} \end{cases}$, and the complementary curves should be constrained by the limits of *OMN* as illustrated in Figure 3. The limits of $E_{Pen}$ and $E_{PT}$ on $E$ can be approximately

satisfied by the sigmoid function H2017 with the parameters transformed from the linear AA function (Han and Tian, 2018a). Besides, adopts $dy_H/dx_H = 0$ as $y_H \rightarrow 1$ by considering that $E$ approaches $E_{Pen}$ under wet conditions, which results in a sigmoid type function.

In the manner of the AA approach of formulating $E_{po}$ and $E_{pa}$, B15 evolves to one of its analytical forms, the polynomial B2015. Taking the Priestley-Taylor coefficient as a parameter, B2015 can also be regarded a polynomial

analytical forms of H12 (Table 2), and can be compared with the sigmoid H2017 in the state space ($E_{rad}/E_{Pen}$, $E/E_{Pen}$) (Figure 3). In the polynomial B2015, the limits on the $E$ are specified to $E \leq \alpha E_{rad} \leq E_{Pen}$. In practice, a constant $\alpha$ is widely used, and the polynomial curves of B2015 are required to be constrained by the triangle domain *OMP* (Figure 3), which discards the domain out of *OMP*. However, the Priestley-Taylor coefficient varies with several factors, such as the relative transport efficiency of turbulent, or the surface/air temperature (Assouline et al., 2016; Szilagyi, 2014). Thus,

$E_{rad}/E_{Pen}$ may be larger than $1/\alpha$, revealing that the trapezoidal domain adopted by the sigmoid H2017 is more accurate. In

the state space ( $E_{rad}/E_{Pen}$ , $E/E_{Pen}$ ), the curve of the sigmoid H2017 exhibits a three-stage pattern, whereas the linear AA and polynomial B2015 have one and two stages respectively. As it is difficult for one site to cover all the three stages with a wide range of wetness, the linear AA can effectively represent the complementary curve under normal conditions falling in the middle stage. The polynomial B2015 is effective if the first two stages exist. Given that the third stage is uncommon, the

380 polynomial B2015 performs well with calibrated parameters (Brutsaert et al., 2017; Han and Tian, 2018a; Liu et al., 2016; Zhang et al., 2017). However, observed points are located in the domain out of *OMP* at several flux sites, and the sigmoid H2017 shows the best performance in estimating evaporation as validated by using data from FLUXNET (Han and Tian, 2018a; Wang et al., 2019).

## 4 Current applications and future developments of the generalized complementary principle

**4.1 Current applications of the generalized complementary functions for evaporation estimation**

Morton (1983) thought that the ability of the complementary principle to estimate actual evaporation by using meteorological variables only can significantly influence the science and practice of hydrology. However, the attempts in using the complementary principle for evaporation estimation in hydrological modelling (Barr et al., 1997; Nandagiri, 2007; Oudin et al., 2005) have been suspended, while those attempts in applying such principle in drought assessment (Hobbins et

al., 2016; Kim and Rhee, 2016) are still in their infancy. Moreover, the potential applications in agriculture water management are limited in the sense that the irrigation-induced changes in potential evaporation was mainly evaluated at an annual timescale (Han et al., 2014a; Han et al., 2017; Ozdogan et al., 2006). Apparently, the complementary principle did not develop to its full capacity via the linear CR, which leaves a broad space for applying the generalized complementary functions for evaporation research.

For example, the generalized complementary functions have been validated or applied in evaporation estimation for many sites (Ai et al., 2017; Brutsaert et al., 2017; Crago and Qualls, 2018; Han and Tian, 2018a; Zhang et al., 2017), and several basins in China (Gao et al., 2018; Liu et al., 2016). B2015 was applied to estimate global terrestrial evaporation with calibrated $\alpha$ as a function of aridity index (Brutsaert et al., 2019). The modified Granger's model was also applied for estimating global evaporation with 0.5° spatial resolution and monthly time steps (Anayah and Kaluarachchi, 2019). It

should be noted that most, if not all, above mentioned CR applications need "prior" knowledge in *E* (either ground-measured or water-balance-derived) to calibrate the parameters. Recently, Szilagyi et al. (2017)'s model was applied for monthly evaporation estimation without calibration across the conterminous China (Ma et al., 2019) and United States (Ma and Szilagyi, 2019). A wide range of model evaluations against the plot-scale flux measurements and basin-scale water balance results suggested that the generalized complementary functions could serve as a benchmark tool for validating the large-scale

*E* results simulated by those Land Surface models and Remote Sensing models (Ma and Szilagyi, 2019). However, further

applications over the world are still needed to develop more long-term, high-resolution $E$ datasets for use in hydrological and atmospheric communities.

## 4.2 Parameterizing generalized complementary functions for future applications

Determining the parameters of the generalized complementary functions is the urgent work for the application of B2015 and H2017 for evaporation estimation, as well as the development of the generalized complementary principle. Given the variations in $\alpha$, the linear AA, polynomial B2015 and sigmoid H2012 all have two parameters. The linear AA with a default value of $b=1$ was applied at first in evaporation estimation. For the B2015, $c$ was thought to be only applied to accommodate unusual situations (Brutsaert, 2015). In practice, $c = 0$ is adopted and the Priestley-Taylor coefficient is calibrated (Brutsaert, 2015; Brutsaert et al., 2017; Liu et al., 2016; Zhang et al., 2017). But the calibrated $\alpha$ is smaller than the widely accepted constant 1.26 or even smaller than the unit at several sites, which is physically unrealistic. Han and Tian (2018a) found that $c$ corresponds to $b$ in the AA by setting the B2015 approximately equal to the AA in the middle stage. However, the default value of $c=0$ corresponds to $b$ with a value around 4.5, not the early default value of $b=1$, implying the default value of $c=0$ may be not suitable. By calibrating both $\alpha$ and $c$, the B2015 performed well in estimating evaporation for 20 FLUXNET sites, and the value of $\alpha$ were more rational (Han and Tian, 2018a).

By contrast, two more parameters ($x_{min}$ and $x_{max}$) are added to the sigmoid H2017. Because the sigmoid complementary curve are insensitive to $x_{min}$ and $x_{max}$, Han and Tian (2018a) suggested that they could be treated as constant parameters for application convenience. $x_{min}$ and $x_{max}$ may change along with $E_{rad}$, and were thought to vary with the time scales (Han and Tian, 2018a), $x_{min} = 0$ and $x_{max} = 1$ are appropriate at a daily scale for convenience, as have be evidenced by the well performances when compared to the flux measurements (Han et al., 2012; Han and Tian, 2018a). $x_{min}$ and $x_{max}$ are expected to be calculated by applying certain approaches to reduce the number of parameters of H2017 to two (Han and Tian, 2019).

Although $\alpha$ would vary in theory (Assouline et al., 2016), it is widely used with a constant value of 1.26 in practice (Priestley and Taylor, 1972). After calibrating, the variations of $\alpha$ is much less significant than those of the other parameters. Moreover, the calibrated $\alpha$ approaches 1.26, especially for the sigmoid H2017. Thus, the constant $\alpha = 1.26$ was suggested with acceptable weakening of the accuracy of $E$ estimation (Han et al., 2012; Han and Tian, 2018a). In practice, $\alpha$ was also determined from the observed $E$ values in wet condition when $E$ is close to $E_{Pen}$ and/or $E_{PT}$ (Kahler and Brutsaert, 2006; Ma et al., 2015a; Wang et al., 2019). A novel method by using observed air temperature and humidity data under wet environment was proposed by Szilagyi et al. (2017) when measured $E$ is lacking, and was successfully used for large-scale CR model applications (Ma and Szilagyi, 2019; Ma et al., 2019).

After determining $\alpha$ in advance, only a single parameter in the generalized complementary functions needs to be calibrated. As the parameters of the B2015 and H2017 can be transferred from the asymmetric parameter $b$ of the original CR (Han and Tian, 2018a), the former studies on the characteristics of $b$ could help its parameterization. The $b$ in the desert was much smaller than those in the oases or irrigated farmlands (Han et al., 2008, 2012). $b$ was thought to be related to the characteristics of the atmosphere, i.e., the atmospheric humidity (Szilagyi, 2015), the Clausius–Clapeyron relationship between saturation-specific humidity and temperature (Lintner et al., 2015), or the characteristics of the land surface, i.e., the surface temperature (Szilagyi, 2007), the water availability of the land surface (Han and Tian, 2018b; 2010), or the ecosystem types (Wang et al., 2019). Szilagyi (2015) applied a sigmoid function of relative humidity to parameterize $b^{-1}$. Wang et al. (2019) used the ecosystem mean $b$ values of 217 sites around the world in the B2017 with litter weakening of the evaporation estimation accuracy. However, the characteristics and determination methods of $b$ need further studies toward a calibration-free evaporation estimation model.

## 4.3 Integrating with other approaches for further development

Actual evaporation is widely estimated as a reduction of the evaporation demand. The reduction factor was first taken as a function of soil moisture (Penman, 1950; Shuttleworth, 1993), or canopy resistance (Monteith, 1965). This Penman approach or Penman-Monteith approach has played a great role in parameterizing the evaporation process in hydrological models and the land surface models. The canopy or surface temperature has also been widely used as a water stress indicator (Jackson et al., 1981; Jackson et al., 1988), and the approach based on land surface temperature from remote sensing data has generated increasing attention. At the annual or long term time scales, the reduction factor is taken as a function of the humidity index represented by the ratio of precipitation to potential evaporation, and this method is known as Budyko approach (Budyko, 1974; Yang et al., 2006; Zhang et al., 2001). In the above approaches, the evaporation demand is assumed to be independent of the land surface (Lhomme, 1997c; Morton, 1983). But at a large area where the land surface significantly interacts with the atmosphere, the evaporation demand will be altered by the changes of the land surface and the independent assumption does not hold. Although problems may not arise in diagnostic modelling as current evaporation demand can be observed, they should be considered if these approaches are applied to a large area and used for future prediction or management in prognostic modelling (Han and Tian, 2018b).

Compared to the above approaches relied on the land surface properties, the reduction factor is determined from the atmospheric wetness in the generalized complementary functions (Table 3). The changes in evaporation demand due to the land surface properties are conceptually considered in the complementary principle, which is a theoretical improvement and would be helpful in predicting evaporation with land use changes. In addition, under the conditions that the land surface properties are difficult to get, it is an obvious advantage of the complementary principle using the routinely observed meteorological variables in evaporation estimation. However, the complementary principle assumes that the changes in land surface properties can be accurately and timely detected from the changes of the atmospheric conditions. This assumption requires that the effects of regional or large-scale advections are negligible (Morton, 1983). Outside these situations, the

generalized complementary functions may not work well because land surface properties are inadequately involved. Besides, the components of evaporation from different patches of the spatially heterogeneous surfaces, especially the evaporation from bare soil and the transpiration from vegetation, cannot be separated in the complementary principle, which is its disadvantage compared to the other approaches.

Considering the above disadvantages, Han and Tian (2018b) proposed a framework to integrate the complementary principle with other approaches for the advancement of evaporation research, which expresses $E/E_{Pen}$ as a function of both the land surface properties and the atmospheric wetness. Actually, both the land surface characteristics (e.g., soil moisture and vegetation) and atmospheric variables (e.g., radiation, humidity, and temperature) have been used in the Jarvis–Stewart model (Jarvis, 1976; Stewart, 1988) to parameterize the canopy resistance. In fact, several attempts were conducted by integrating the complementary principle with other approaches to derive some of the land surface variables by using the meteorological variables (Han et al., 2015; Mallick et al., 2013; Szilagyi and Jozsa, 2009). A unified formulation of Penman approach and the linear AA function was proposed by Crago and Brutsaert (1992). The integrated approach is a more rational conceptualization of the evaporation process from the unsaturated surface into the unsaturated atmosphere, and is expected to increase the accuracy of evaporation estimation while reducing the burdens of parameterization. The findings of Liu et al. (2018) and Wang et al. (2019) that the parameters of the generalized functions significantly depends on the wetness of the land surface have demonstrated that the integrated approach has a bright prospect. However, proper manners to integrate them need further studies.

**5 Conclusions**

The complementary principle conceptualizes the feedbacks of land surface evaporation on atmospheric evaporation demand and offers advantages in evaporation estimation. In this study, the historical development of the complementary principle during the past half century was reviewed and the two types of generalized complementary functions were focused. In addition, future development for the generalized complementary principle was summarized based on the review. The concluding remarks are as follows:

(1) The studies on the complementary principle adopted a symmetric CR at first, and then extended to an asymmetric CR. At present the original CR has evolved to the generalized complementary principle, which employs nonlinear functions as generalizations of the original linear relationship. The generalized complementary principle has a more rigorous physical base and offers potential in advancing actual evaporation estimation by using simple and standardized procedures.

(2) Two types of generalized complementary functions were derived based on different understandings of the boundary conditions under completely wet environments: the sigmoid H12 and polynomial B15. The B15 inherits the concepts of "potential evaporation $E_{po}$" and "apparent potential evaporation $E_{pa}$" from the original CR, and uses a polynomial function relating $E/E_{pa}$ to $E_{po}/E_{pa}$. By contrast, H12 goes further from the original CR without involving the difficulties in formulating $E_{po}$ and $E_{pa}$. Instead, a sigmoid function relating the ratio of actual evaporation to the Penman potential

evaporation $E_{\text{Pen}}$ and the proportion of the radiation component in $E_{\text{Pen}}$ was derived. Nevertheless, further validation and application of the two types of generalized complementary functions are required with multiple dataset from different parts of the world.

(3) Further studies from both the theoretical and practical aspects are still required before the generalized complementary principle achieves its potential. The generalized complementary principle requires a bold attempt for the practice of hydrology through enhancing its ability of evaporation estimatation while reducing the burdens of parameterization. Thus, it should be carefully examined for its physical base of the boundary conditions under completely wet environment, and be integrated with other approaches to include the information of the land surface properly.

## Appendix: List of symbols

| | | |
|---|---|---|
| Abbreviations of complementary functions | AA | Advection-aridity function proposed by Brutsaert and Stricker (1979) |
| | H12 | Generalized complementary function proposed by Han et al., (2012) |
| | H2012 | Sigmoid analytical form of H12 proposed by Han et al., (2012) |
| | H2017 | Sigmoid analytical form of H12 proposed by Han and Tian (2018) |
| | B15 | Generalized complementary function proposed by Brutsaert (2015) |
| | B2015 | Polynomial applicable form of B15 suggested by Brutsaert (2015) |
| | C2016 | Rescaled applicable form of B15 proposed by Crago et al., (2016) |
| | S2017 | Rescaled applicable form of B15 proposed by Szilagyi et al., (2017) |
| Three types of evaporation in CR | $E$ | Actual evaporation |
| | $E_{pa}$ | Apparent potential evaporation in CR |
| | $E_{po}$ | Potential evaporation in CR |
| Specific formulations for $E_{pa}$ or $E_{po}$ | $E_{Pan}$ | Pan evaporation |
| | $E_{Pen}$ | Penman's potential evaporation (Penman, 1948) |
| | $E_{rad}$ | Radiation term of $E_{Pen}$ |
| | $E_{aero}$ | Aerodynamic term of $E_{Pen}$ |
| | $E_{Pen}^{KP}$ | Modified Penman's equation by Kohler and Parmele (1967) |
| | $E_{Pen}^{r_s}$ | Penman's potential evaporation with temperature and humidity calculated from the ABL model corresponding to certain surface resistance ($rs$) |
| | $E_{Pen}^{r_s=0}$ | Penman's potential evaporation with temperature and humidity calculated from the ABL model corresponding to $rs=0$ |
| | $E_{PM}^{r_s\min}$ | Penman–Monteith (Monteith, 1965) evaporation with a minimum surface resistance |
| | $ET_0$ | Reference crop evapotranspiration (Allen et al., 1998) |
| | $E_{MT}$ | Mass-transfer type potential evaporation (van Bavel, 1966) |
| | $E_{Mor}$ | Morton (1983)'s potential evaporation |
| | $E_{PT}$ | Priestley-Taylor's(Priestley and Taylor, 1972) minimal advection evaporation |
| | $E_{PT}^{T_p}$ | Morton's modified Priestley-Taylor's minimal advection evaporation (Morton, 1983) |
| | $E_{PT}^{T_{ws}}$ | Szilagyi and Jozsa (2008)'s modified Priestley-Taylor's minimal advection evaporation |
| | $E_{Pen}^{\max}$ | Maximum value of $E_{pa}$ calculated by Penman equation (Szilagyi et al., 2017) |
| | $E_{MT}^{\max}$ | Maximum value of $E_{pa}$ calculated by a mass transfer approach (Crago et al., 2016) |
| Parameters in CR | $\alpha$ | Priestley-Taylor coefficient |
| | $b$ | Symmetry parameter of the CR |
| Meteorological variables used for calculating $E_{pa}$ or $E_{po}$ | $T_a$ | Air temperature |
| | $T_s$ | Surface temperature |
| | $T_{ws}$ | Surface temperature under wet environment defined by Szilagyi and Jozsa (2008) |
| | $T_p$ | Equilibrium temperature defined by Morton (1983) |
| | $\Delta$ | Slope of the saturation vapor curve |
| | $\gamma$ | Psychrometric constant |
| | $R_n$ | Net radiation |
| | $G$ | Ground heat flux |

| *RH* | Relative humidity |
|------|-------------------|

**Acknowledgments**

This research was partially sponsored by the National Natural Science Foundation of China (No. 51579249, 51825902), the National Key Research and Development Program of China (No. 2016YFC0402707), the Research Fund (No. 2016ZY06) of State Key Laboratory of Simulation and Regulation of Water Cycle in River Basin, China Institute of Water Resources and Hydropower Research.

Code and data availability. There is no code and data used in this review paper.

Author contributions. SH and FT jointly developed the review and edited the manuscript. SH drafted the paper.

Competing interests. The authors have declared no conflicts of interest for this article.

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
