# Peer review of "A review of complementary principle of evaporation: From original linear relationship to generalized nonlinear functions"

_Hydrology and Earth System Sciences, 2019_

## Referee Comment (RC1) · Anonymous Referee #1 · 31 Dec 2019

The paper summarizes the history of the complementary principle of evaporation, with an emphasis on how the symmetric linear complementary relationship develops to generalized nonlinear functions. In general, I enjoyed reading such a review on the CR, which was widely used to estimate ET over different spatial and/or temporal scales.

My main comment is there are still some works worth being discussed, though this review is overall complete: 1) The perspective of Lhomme and Guilioni (2006, 2010) which relates potential evaporation to surface resistance. 2) Aminzadeh et al. (2016)'s CR with Ep defined by a surface temperature

Also, there are a few latest CR studies in 2019 that are highly relevant to the submitted

manuscript, e.g., Anayah & Kaluarachchi (2019) and Brutsaert et al. (2019). Could the authors discussed a little bit?

Specific Comments: Line 19: Is the boundary condition here specified to the wet environment?

Line 40: Recent publications using GCR in 2019 for estimating evaporation should be added here.

Line 87: "while" should be replaced by "whereas"?

Line 107: the "realistic" is compared to the former model. I think adding "more" here may be better.

Line 110: "wss"??

Line 167: "The asymmetric CR is widely used", please revised this sentence

Line 179: More statements on the asymmetric CR should be added, including the negative relationship between E/Epo and Epa/Epo was treated as an extension of the original CR, and the validation in several locations.

Line 246: "Han and Tian (2018) further validated the sigmoid feature": Please state the work more detailed because there are still controversies on it.

Line 270: What is the essential difference between B15 and H12? Is "B15 inherits all three types of evaporation dated from the original CR"? Please rearrange these sentences.

Line 304: The varying characteristics of the PT coefficient should be introduced here

Line 359: Brutsaert's recent work by using c=0 and varying PT coefficient should be added. Check Brutsaert et al. (2019). The Conclusion part could be improved. I wonder are there any outlooks for future studies on CR could be summarized using a few sentences here?

Reference: Aminzadeh, M., Roderick, M. L., & Or, D. (2016). A generalized complementary relationship between actual and potential evaporation defined by a reference surface temperature. Water Resources Research, 52(1), 385-406. doi:10.1002/2015wr017969 Anayah, F. M., & Kaluarachchi, J. J. (2019). Estimating Global Distribution of Evapotranspiration and Water Balance Using Complementary Methods. Atmosphere-Ocean, 57(4), 279-294. doi:10.1080/07055900.2019.1656052 Brutsaert, W., Cheng, L., & Zhang, L. (2019). Spatial Distribution of Global Landscape Evaporation in the Early Twenty First Century by Means of a Generalized Complementary Approach. Journal of Hydrometeorology. doi:10.1175/jhm-d-19-0208.1 Lhomme, J. P., & Guilioni, L. (2006). Comments on some articles about the complementary relationship. Journal of Hydrology, 323, 1-3. Lhomme, J. P., & Guilioni, L. (2010). On the link between potential evaporation and regional evaporation from a CBL perspective. Theoretical and Applied Climatology, 101(1), 143-147.
* * *

---

## Referee Comment (RC2) · Anonymous Referee #2 · 1 Jan 2020

The authors present a very detailed review on the studies and developments of the complementary relationship of evaporation. Although the review is very detailed and scientifically well supported on the existing literature, I think it is too heavy due to the load of parameters introduced and unexplained, the long list of studies mentioned and a weak coherency when enumerating the studies. Can the authors make this easier for the reader to read through?

Furthermore, I am completely missing the incites and perspectives from the authors. It would be very nice to see the opinion from the authors regarding the benefits of the framework. I suggest an extra section discussing 1) the best approach according

to the authors criteria, 2) the future of the CR for E estimation, and 3) a comparison highlighting the advantages, disadvantages and opportunities of using the CR principle against other methods of Evaporation estimation that are not mentioned here. After this, the review should be ready for publication.

L. 24 State if it is a positive or negative feedback.

L. 27 To understand, so the differences between Epa and Epo is just that Epa is small and local and Epo large-scale? Can you provide more explanation on what these two variables really mean since they are so important for this discussion? Specially for understanding Figure 1.

L. 32 What complex formulations?

L. 35-39 Please rephrase, it is difficult to understand.

L. 38 If there is a Penman calculation variable Epen, then how do you estimate Epa and Epo, that is different from Penman. Please specify.

Table 1. Nice table!, but refer to the Appendix for the unexplained parameters.

L. 69 why "basin-wide water balance" results? You said before that Epa is from a "small saturated surface".

L. 70-75 But have these estimates been validated in some way?

L. 75 When they found that it is overestimating or underestimating E, how did these studies obtain the real E then?

L. 78 IMPORTANT. Since you are constantly introducing many parameters related to actual or potential evaporation, please include in the appendix a detailed explanation on the difference between each E parameter. For instance, to know how Epan differs from Epa.

L. 88 Do you mean that they change in opposite directions with increasing water availability? L95 "the governing changes"

L. 98 Why does Morton say that it is unrealistic and does not have proof, and argue against it, since you are performing a review on the subject.

L. 113 You mention an asymmetry, but before you were talking about symmetry?

L. 136 So when is $E\_PT$ different from Epo. In other words, more clarity between these terms.

L. 135-156 IMPORTANT I find these paragraphs hard to read and somehow "boring". As in a review, it would be very good if you can try to articulate all the studies in a more consistent way so that it does not become a list of studies and references each with a brief explanation. Also, many, many terms that have not been previously explained, only in an appendix. As it is, the review paper is now more focus to experts in the CR that common hydrologists.

Section 3. I don't see the rationale behind the selection of the subtitles 3.1 and 3.2. A brief explanation is needed. Why these subtitles, I assume 3.1 are the symmetric approaches and 3.2 the asymmetric ones? Think on the reader that is reading this review.

L.164 so b=1 means symmetry?

L. 167. "The asymmetric CR is widely used?"

Can you make a paragraph saying in your point of view which approach is better and why, symmetric or asymmetric?

Can you make a similar Table 1 but for the non-linear relationships? I think that you mention many approaches that are not included in Table 2.
* * *

---

## Referee Comment (RC3) · Anonymous Referee #3 · 8 Jan 2020

The authors provided an informative yet in-depth review of the research activities over the past half-century using the complementary principle of evaporation. I enjoyed the reading and feel it is overall a timely and nice contribution to the hydrology community. nevertheless, I do challenge the authors to elevate the presentation quality, in a spirit to make it valuable for a variety of audiences, from those who are not very familiar with this type of research but would like to have some background information, to those who are really active in the field. In its current form, the writing style is still, more or less, in favor of the latter. Also, I feel it might be more straightforward to include the word "review" in the title and abstract. Otherwise, some readers may not realize this is a review article until the end of the introduction.

---

## Author Comment (AC1) · 10 Feb 2020

**Response to Referee #1**

**General Comments**

My main comment is there are still some works worth being discussed, though this review is overall complete: 1) The perspective of Lhomme and Guilioni (2006, 2010) which relates potential evaporation to surface resistance. 2) Aminzadeh et al. (2016)'s CR with Ep defined by a surface temperature Also, there are a few latest CR studies in 2019 that are highly relevant to the submitted manuscript, e.g., Anayah & Kaluarachchi (2019) and Brutsaert et al. (2019). Could the authors discussed a little bit?

Response: Thanks for the Referee's suggestions. We will discuss these publications in the revised manuscript.

In Lhomme and Guilioni (2006, 2010)'s perspective of CR, the surface resistance is related to the equilibrium evaporation and potential evaporation. We found that it is a linear function relating $E/E_{Pen}$ to $E_{rad}/E_{Pen}$ without intercept in the normalized form. Thus, we will add following paragraph to Section "Normalized complementary functions" in the revised manuscript: "Based on the examination of the CR using a model of the convective boundary-layer with entrainment (Lhomme, 1997), Lhomme and Guilioni (2010, 2006) recommended a form of CR through the effective surface resistance of the region. Integrating this CR into Penman–Monteith equation and the normalization by $E_{Pen}$ lead to

$$\frac{E}{E_{Pen}} = (1+\omega)\frac{E_{rad}}{E_{Pen}} , \tag{1}$$

where $\omega$ is a positive coefficient accounting for the entrainment of dry air within the ABL. Equation (1) is a linear function without intercept, but was not verified and applied using observed data."

Aminzadeh et al. (2016) derived a steady state surface temperature via the surface energy balance at which the sensible heat flux is zero, and calculated $E_{pa}$ and $E_{po}$ using a mass-transfer type reference evaporation corresponding to current and saturated surface water content. We will discuss it in addition to the works of Morton (1983) and Szilagyi and Jozsa (2008) in the revised manuscript.

The latest studies of using CR for global ET estimation will be discussed in the revised manuscript. "B2015 by setting c=0 was applied to estimate global terrestrial evaporation with calibrated $\alpha$ as a function of aridity index (Brutsaert et al., 2019). The modified Granger's model was also applied for estimating global evaporation with 30 min spatial resolution and monthly time steps (Anayah and Kaluarachchi, 2019)."

**Specific Comments:**
Line 19: Is the boundary condition here specified to the wet environment?

    Response: Yes. We will add "under wet environments" after it in the revised manuscript.

Line 40: Recent publications using GCR in 2019 for estimating evaporation should be added here.

    Response: We will add Brutsaert's latest work here.

Brutsaert, W., Cheng, L., and Zhang, L.: Spatial Distribution of Global Landscape Evaporation in the Early Twenty First Century by Means of a Generalized Complementary Approach, J Hydrometeorol, 10.1175/jhm-d-19-0208.1, 2019.

Line 87: "while" should be replaced by "whereas"?

    Response: It will be revised.

Line 107: the "realistic" is compared to the former model. I think adding "more" here may be better.

    Response: "more" will be added here.

Line 110: "wss"??

    Response: We are sorry for the typo. It should be "was"

Line 167: "The asymmetric CR is widely used", please revised this sentence

    Response: We will replace this sentence by "The asymmetric CR is a significant improvement of the symmetric CR, and the opposite changes of $E/E_{po}$ and $E_{pa}/E_{po}$ against $E/E_{pa}$ were treated as an enhanced illustration of the CR (Hu et al., 2018; Zhang et al., 2017; Ma et al., 2015; Brutsaert et al., 2019; Szilagyi, 2007)." and revised this paragraph.

Line 179: More statements on the asymmetric CR should be added, including the negative relationship between E/Epo and Epa/Epo was treated as an extension of the original CR, and the validation in several locations.

    Response: More statements will be added, as "The asymmetric CR is a significant improvement of the symmetric CR, and the opposite changes of $E/E_{po}$ and $E_{pa}/E_{po}$ against $E/E_{pa}$ were treated as an enhanced illustration of the CR (Hu et al., 2018; Zhang et al., 2017; Ma et al., 2015; Brutsaert et al., 2019; Szilagyi, 2007). The performances on evaporation estimation are improved by calibrating the asymmetry parameter $b$ (Kahler and Brutsaert, 2006; Han et al., 2008; Huntington et al., 2011; Ma et al., 2015). Efforts have also made to calculate $b$ by using the meteorological variables, which enhance the predict ability of the CR (Szilagyi, 2015;

Szilagyi, 2007; Aminzadeh et al., 2016). However, the changes in $b$ imply a potential nonlinear characteristic of the CR Han (2008); Lintner et al. (2015). The observed values of $E/E_{po}$ and $E_{pa}/E_{po}$ even exhibit a positive correlation under wet conditions at several flux sites, which challenges the CR (Han and Tian, 2018). But previous studies on the validity of CR have two limitations. First, the true correlation between Epa+ and E+ would be masked when they are both plotted against moisture index (Pettijohn and Salvucci, 2009; Lintner et al., 2015). Second, the wet conditions where the two curves of $E/E_{po}$ and $E_{pa}/E_{po}$ approach were seldom focused, which may hide the true correlation under wet environments."

Line 246: "Han and Tian (2018) further validated the sigmoid feature": Please state the work more detailed because there are still controversies on it.

    Response: We will revised this sentence as "Han and Tian (2018) further validated the sigmoid feature according to the much larger regression slopes of $E/E_{Pen}$ upon $E_{rad}/E_{Pen}$ in the middle stage than those in the other two stages with smaller or larger values of $E_{rad}/E_{Pen}$ by using 22 eddy covariance towers from the FLUXNET (Baldocchi et al., 2001) dataset which includes representative biomes of grasslands, croplands, shrublands, evergreen needleleaf forests, deciduous broadleaf forests, and wetlands."

Line 270: What is the essential difference between B15 and H12? Is "B15 inherits all three types of evaporation dated from the original CR"? Please rearrange these sentences.

    Response: The two generalized complementary approaches, H12 and B15, are essentially different, with completely different normalized variables (Table 3). B15 inherits the concept of the three types of evaporation dated from the original CR, and its boundary conditions and analytical form are derived for $x_B = E_{po}/E_{pa}$ and $y_B = E/E_{pa}$. By contrast, H12 goes much further from the original CR. The boundary conditions and the analytical form of H12 are derived for $x_H = E_{rad}/E_{Pen}$ and $y_H = E/E_{Pen}$. We will add a new subsection "Comparisons between the two generalized complementary approaches" to discuss the essential difference between B15 and H12.

Line 304: The varying characteristics of the PT coefficient should be introduced here

    Response: We will introduce it here, as "the Priestley-Taylor coefficient varies with several factors, such as the relative transport efficiency of turbulent, or the

surface/air temperature (Assouline et al., 2016; Szilagyi, 2014)."

Line 359: Brutsaert's recent work by using c=0 and varying PT coefficient should be added. Check Brutsaert et al. (2019).

    Response: We will add it here.

The Conclusion part could be improved. I wonder are there any outlooks for future studies on CR could be summarized using a few sentences here?

    Response: We will rearrange the conclusion part and add three points about future studies: 1) Integrating the complementary principle with other approaches for future development; 2) Assessing the generalized complementary functions over varied places with gradient climate and landscape features; and 3) increasing the accuracy of evaporation estimation while reducing the burdens of parameterization.

[revised manuscript text omitted]

---

## Author Comment (AC2) · 10 Feb 2020

**Response to Referee #2**

**General Comments**

The authors present a very detailed review on the studies and developments of the complementary relationship of evaporation. Although the review is very detailed and scientifically well supported on the existing literature, I think it is too heavy due to the load of parameters introduced and unexplained, the long list of studies mentioned and a weak coherency when enumerating the studies. Can the authors make this easier for the reader to read through?

Response: In the revised manuscript, we will rearrange the manuscript, and try to make it easier for the reader to read through. Section "2 Symmetric complementary relationship" and "3 Efforts in maintaining a linear complementary relationship" will be combined to one section "Linear complementary relationship" to make it more coherent. The parameters will be introduced and explained more clearly in the revised manuscript.

Furthermore, I am completely missing the incites and perspectives from the authors. It would be very nice to see the opinion from the authors regarding the benefits of the framework. I suggest an extra section discussing 1) the best approach according to the authors criteria, 2) the future of the CR for E estimation, and 3) a comparison highlighting the advantages, disadvantages and opportunities of using the CR principle against other methods of Evaporation estimation that are not mentioned here. After this, the review should be ready for publication.

Response: In the revised manuscript, we will give a clear point of view on the different approaches: the asymmetric CR is a significant improvement of the symmetric CR, and the generalized **complementary principle via nonlinear functions is the recent development. We will also compare the** two generalized complementary approaches in a new subsection and give our perspectives. We will add a new section discussing the current practice and future development of the CR for *E* estimation, and compare it with other methods (the Penman approach, the Budyko approach and others) on the advantages, disadvantages and opportunities of using the CR principle. At last, we propose a suggestion of integrating these approaches for a new generation of evaporation estimation method.

**Specific Comments**

L. 24 State if it is a positive or negative feedback.

Response: We will revise this sentence as "this principle originated from the negative feedback of areal evaporation on evaporation demand (Bouchet, 1963)".

L. 27 To understand, so the differences between Epa and Epo is just that Epa is small and local and Epo large-scale? Can you provide more explanation on what these two variables really mean since they are so important for this discussion? Specially for understanding Figure 1.

Response: The major differences between Epa and Epo are that they

correspond to different atmosphere characteristics. *Epa* corresponds to the atmosphere in contact with current non-saturated evaporating surface as the overpassing air is not affected by the small saturated surface, whereas the atmosphere corresponding to *Epo* is in contact with the large-scale saturated surface. Thus, the surface water availability can be detected from the relative magnitude of *Epa* and *Epo* (as shown in Figure 1), and *E* can be estimated without the knowledges of the surface. We will provide more explanation in the revised manuscript.

**L. 32 What complex formulations?**

Response: The formulations of *Epa* and *Epo* are introduced in Section 3.1. We will add a tip in the revised manuscript.

L. 35-39 Please rephrase, it is difficult to understand. L. 38 If there is a Penman calculation variable Epen, then how do you estimate Epa and Epo, that is different from Penman. Please specify.

Response: It should be noted that  $E_{pa}$  and  $E_{po}$  are theoretical concepts, and need to be formulated when applying for practical problems. The generalized complementary function comes in two ways. Brutsaert (2015) adopted a polynomial function to describe the relationship between E,  $E_{pa}$  and  $E_{po}$ , and suggested to formulate  $E_{pa}$  and  $E_{po}$  by using Penman's potential evaporation ( $E_{Pen}$ ) and Preistley-Taylor's minimal advection evaporation to formulate, respectively. By contrast, Han and Tian (2018) abandoned the theoretical concept of  $E_{pa}$  and  $E_{po}$ , yet used a sigmoid function to describe the relationship among E, Penman's potential evaporation ( $E_{Pen}$ ), and its radiation term ( $E_{rad}$ ), which can be directly used for practical problems. We will rephrase these sentences to make them clear.

Table 1. Nice table!, but refer to the Appendix for the unexplained parameters.Response: We will add it below the table.

L. 69 why "basin-wide water balance" results? You said before that Epa is from a "small saturated surface".

Response: Epa was derived from a hypothetical theoretical concept of "small saturated surface", which means that the "small saturated surface" does not affect the atmosphere, and Epa is determined by the atmosphere corresponding to current unsaturated surface. In application, Epa is calculated by using the meteorological variables corresponding to current unsaturated surface, and is used to calculate the basin-wide actual evaporation. In the revised manuscript, we will make the statement more clear.

L. 70-75 But have these estimates been validated in some way?

Response: Yes. We will introduce the validation of the AA approach "has been validated based on hourly (Parlange and Katul, 1992; Crago and Crowley, 2005), daily (Brutsaert and Stricker, 1979; Ali and Mawdsley, 1987; Qualls and Gultekin, 1997), monthly (Xu and Singh, 2005; Lemeur and Zhang, 1990; Hobbins et al., 2001), and annual (Ramirez et al., 2005; Yu et al., 2009) data from either plot-scale lysimeters and eddy-covariance measurements or basin-wide water balance-derived results."

L. 75 When they found that it is overestimating or underestimating E, how did these studies obtain the real E then?

Response: In the revised manuscript, we will add the potential causes of the

bias "imperfect formulations of  $E_{pa}$  and/or  $E_{po}$ , external energy sources, or even the

nonlinear nature of the complementary principle were considered as potential causes of this bias (Qualls and Gultekin, 1997; Hobbins et al., 2001; Han et al., 2008, 2012)." Please refer to Section "2.4 Efforts in maintaining a linear complementary relationship through rational formulation of  $E_{pa}$  and/or  $E_{po}$ ", and "3.2 Sigmoid function relating  $E/E_{Pen}$  to  $E_{rad}/E_{Pen}$ " for the methods to obtain the real E.

L. 78 IMPORTANT. Since you are constantly introducing many parameters related to actual or potential evaporation, please include in the appendix a detailed explanation on the difference between each E parameter. For instance, to know how Epan differs from Epa.

Response: Generally speaking,  $E_{pa}$  and  $E_{po}$  are theoretical concepts, whereas Epan, Epen, Ept and others are the specifications of them. We will explain it in the appendix.

L. 88 Do you mean that they change in opposite directions with increasing water availability?

Response: Yes. We will change "while" to "whereas" to make the sentence more clear.

L95 "the governing changes"

Response: We will change it in the revised manuscript.

L. 98 Why does Morton say that it is unrealistic and does not have proof, and argue against it, since you are performing a review on the subject.

Response: Morton derived the CR by two assumptions: the net radiation will not change with the surface, and the heat and vapor eddy transfer characteristics are

identical for E and  $E_{pa}$ . Szilagyi (2001) relaxed the second assumption of Morton

(1983). LeDrew (1979) argued that Morton's two assumption do not necessarily hold, and pointed out that the symmetric CR is physically unrealistic by using a diagnostic

model of the energy fluxes within a closed system. We will rephrase these sentences in the revised manuscript.

L. 113 You mention an asymmetry, but before you were talking about symmetry? Response: It should be "symmetric". We are sorry for the typo.

L. 136 So when is E\_PT different from Epo. In other words, more clarity between these terms.

Response: In theory,  $E_{po}$  is the theoretical potential evaporation when the land

surface is saturated, and should be calculated with a proper formula by using "potential" meteorological variables corresponding to the saturated surface. The Priestley-Taylor equation has been widely accepted to represent evaporation from extensive saturated surfaces, by using meteorological variables corresponding to these saturated surfaces (Brutsaert, 1982; Priestley and Taylor, 1972). This way it was

suggested to represent  $E_{po}$  (Brutsaert and Stricker, 1979). We will make the

statement more clear in the revised manuscript.

L. 135-156 IMPORTANT I find these paragraphs hard to read and somehow "boring". As in a review, it would be very good if you can try to articulate all the studies in a more consistent way so that it does not become a list of studies and references each with a brief explanation. Also, many, many terms that have not been previously explained, only in an appendix. As it is, the review paper is now more focus to experts in the CR that common hydrologists.

Response: We will rephrase this paragraph to make is more consistent and easy to read. In the first paragraph, the first problem of using EPT to denote Epo is pointed, and is explained as "predicting the hypothetical surface or air temperature

corresponding to the extensive saturated surface is critical for rational defining  $E_{po}$ ".

Then, three works aiming to settle this problem are introduced one by one: Morton (1983), Szilagyi and Jozsa (2008), and Aminzadeh et al. (2016).

Next, another problem is pointed as "Advection is another factor influencing

 $E_{po}$ . However,  $E_{PT}$  does not fully consider the effects of advection, which are

inevitable in reality (Morton, 1983, 1975; Parlange and Katul, 1992)." The works of Morton (1983) and Parlange and Katul (1992) are introduced.

Section 3. I don't see the rationale behind the selection of the subtitles 3.1 and 3.2. A brief explanation is needed. Why these subtitles, I assume 3.1 are the symmetric approaches and 3.2 the asymmetric ones? Think on the reader that is reading this review.

Response: We will combine section 2 and 3 and change the order of former subsection 3.1 and 3.2. We believe it will be more rationale following this order:

2.1 Concept of symmetric complementary relationship;

2.2 Proofs of symmetric CR;

2.3 Asymmetric linear CR as an extension;

2.4 Efforts in maintaining a linear complementary relationship through.

The new 2.4 is for both the symmetric and asymmetric CR.

L.164 so b=1 means symmetry?

Response: Yes. We will add this in the revised manuscript.

L. 167. "The asymmetric CR is widely used?" Can you make a paragraph saying in your point of view which approach is better and why, symmetric or asymmetric?

Response: We will delete this sentence and add a paragraph as:

The asymmetric CR is a significant improvement of the symmetric CR, and the opposite changes of  $E/E_{po}$  and  $E_{pa}/E_{po}$  against  $E/E_{pa}$  were treated as an enhanced illustration of the CR (Hu et al., 2018; Zhang et al., 2017; Ma et al., 2015; Brutsaert et al., 2019; Szilagyi, 2007). The performances on evaporation estimation are improved by calibrating the asymmetry parameter b (Kahler and Brutsaert, 2006; Han et al., 2008; Huntington et al., 2011; Ma et al., 2015). Efforts have also made to calculate b by using the meteorological variables, which enhance the predict ability of the CR (Szilagyi, 2015; Szilagyi, 2007; Aminzadeh et al., 2016). However, the changes in b imply a potential nonlinear characteristic of the CR Han (2008); Lintner et al. (2015). The observed values of  $E/E_{po}$  and  $E_{pa}/E_{po}$  even exhibit a positive correlation under wet conditions at several flux sites, which challenges the CR (Han and Tian, 2018). But previous studies on the validity of CR have two limitations. First, the true correlation between Epa+ and E+ would be masked when they are both plotted against moisture index (Pettijohn and Salvucci, 2009; Lintner et al., 2015). Second, the wet conditions where the two curves of  $E/E_{po}$  and  $E_{pa}/E_{po}$  approach were seldom focused, which may hide the true correlation under wet environments.

Can you make a similar Table 1 but for the non-linear relationships? I think that you mention many approaches that are not included in Table 2.

Response: We will make a Table for the nonlinear generalized complementary functions as:

| Туре    | Formula *                                         | References                                        |
|---------|--------------------------------------------------------------|---------------------------------------------------|
| Linear  | $y = \alpha(1 + \frac{1}{b})x - \frac{1}{b}$                 | Brutsaert and Stricker (1979)                     |
|         | $y = (1 + \omega)x$                                          | Lhomme and Guilioni (2010, 2006)                  |
| Sigmoid | $y = \frac{1}{1 + c_1 e^{d(1 - x)}}$                         | Granger (1989), Han et al.
(2011)              |
|         | $y = \frac{1}{1 + m(\frac{1}{x} - 1)^n}$                     | Han et al. (2012)                                 |
|         | $y = \frac{1}{1 + m(\frac{x_{\max} - x}{x - x_{\min}})^n}$   | Han and Tian (2018)                               |
| Concave | $y = \frac{1}{1 + k(\frac{1}{x} - 1) + l}$                   | Katerji and Perrier (1983),
Han et al. (2014b) |
|         | $y = (2-c)\alpha^2 x^2 - (1-2c)\alpha^3 x^3 - c\alpha^4 x^4$ | Brutsaert (2015)                                  |

Table 2. Different formulas for normalized complementary functions

[revised manuscript text omitted]

---

## Author Comment (AC3) · 10 Feb 2020

The authors provided an informative yet in-depth review of the research activities over the past half-century using the complementary principle of evaporation. I enjoyed the reading and feel it is overall a timely and nice contribution to the hydrology community. Nevertheless, I do challenge the authors to elevate the presentation quality, in a spirit to make it valuable for a variety of audiences, from those who are not very familiar with this type of research but would like to have some background information, to those who are really active in the field. In its current form, the writing style is still, more or less, in favor of the latter.

Response: Thanks for the positive comment to our manuscript. In the revised manuscript, we will add a new section "Integrating with other approaches for future development" to compare the complementary approach of evaporation estimation with the approaches which are more popular in the research community, such as the Penman approach, the Budyko approach et., and propose a suggestion of integrating these approaches with the complementary principle for a new generation of evaporation estimation method. We think this new section would help the readers who are not very familiar with the complementary principle. In addition, we will try to elevate the presentation quality.

Also, I feel it might be more straightforward to include the word "review" in the title and abstract. Otherwise, some readers may not realize this is a review article until the end of the introduction.

Response: We will change the title to "A review of complementary principle of evaporation: From original linear relationship to generalized nonlinear functions", and the word "review" will also be included in the abstract.

---

## Author Response (AR1)

**Response to the referees**

The authors gratefully thank to the editor and referees for their critical comments on our manuscript which drives us to improve the manuscript greatly. The comments and questions were addressed point by point.

**Response to Referee #1**

**General Comments**

My main comment is there are still some works worth being discussed, though this review is overall complete: 1) The perspective of Lhomme and Guilioni (2006, 2010) which relates potential evaporation to surface resistance. 2) Aminzadeh et al. (2016)'s CR with Ep defined by a surface temperature Also, there are a few latest CR studies in 2019 that are highly relevant to the submitted manuscript, e.g., Anayah & Kaluarachchi (2019) and Brutsaert et al. (2019). Could the authors discussed a little bit?

Response: Thanks for the Referee's suggestions. We discussed these publications in the revised manuscript.

In Lhomme and Guilioni (2006, 2010)'s perspective of CR, the surface resistance is related to the equilibrium evaporation and Penman's potential evaporation. We found that it is a linear function relating $E/E_{Pen}$ to $E_{rad}/E_{Pen}$ without intercept in the normalized form. Thus, we added following paragraph to Section "Normalized complementary functions". Please refer to Lines 216-221 of the revised manuscript.

Aminzadeh et al. (2016) derived a steady state surface temperature via the surface energy balance at which the sensible heat flux is zero, and calculated $E_{pa}$ and $E_{po}$ using a mass-transfer type reference evaporation corresponding to current and saturated surface water content. We discussed it in addition to the works of Morton (1983) and Szilagyi and Jozsa (2008) in the revised manuscript (Lines 193-195)

The latest studies of using CR for global ET estimation is discussed in the revised manuscript as "The polynomial B2015 was applied to estimate global terrestrial evaporation with calibrated $\alpha$ as a function of aridity index (Brutsaert et al., 2019). The modified Granger's model was also applied for estimating global evaporation with 30 min spatial resolution and monthly time steps (Anayah and Kaluarachchi, 2019)."

**Specific Comments**

Line 19: Is the boundary condition here specified to the wet environment?

Response: Yes. We added "under wet environments" after it in the revised manuscript.

Line 40: Recent publications using GCR in 2019 for estimating evaporation should be

added here.

Response: We added Brutsaert's latest work here.

Brutsaert, W., Cheng, L., and Zhang, L.: Spatial Distribution of Global Landscape Evaporation in the Early Twenty First Century by Means of a Generalized Complementary Approach, J Hydrometeorol, 10.1175/jhm-d-19-0208.1, 2019.

Line 87: "while" should be replaced by "whereas"?

Response: It was revised.

Line 107: the "realistic" is compared to the former model. I think adding "more" here may be better.

Response: "more" was added here.

Line 110: "wss"??

Response: We are sorry for the typo. It should be "was"

Line 167: "The asymmetric CR is widely used", please revised this sentence

Response: We replaced this sentence by "The asymmetric CR is a significant improvement of the symmetric CR, and the opposite changes of $E/E_{po}$ and $E_{pa}/E_{po}$ against $E/E_{pa}$ were treated as an enhanced illustration of the CR (Hu et al., 2018; Zhang et al., 2017; Ma et al., 2015; Brutsaert et al., 2019; Szilagyi, 2007)."

Line 179: More statements on the asymmetric CR should be added, including the negative relationship between E/Epo and Epa/Epo was treated as an extension of the original CR, and the validation in several locations.

Response: More statements on the asymmetric CR were added as "The asymmetric CR is a significant improvement of the symmetric CR, and the opposite changes of $E/E_{po}$ and $E_{pa}/E_{po}$ against $E/E_{pa}$ were treated as an enhanced illustration of the CR (Hu et al., 2018; Zhang et al., 2017; Ma et al., 2015; Brutsaert et al., 2019; Szilagyi, 2007). The performances on evaporation estimation are improved by calibrating the asymmetry parameter b (Kahler and Brutsaert, 2006; Han et al., 2008; Huntington et al., 2011; Ma et al., 2015). Efforts have also made to calculate b by using the meteorological variables, which enhance the predict ability of the CR (Szilagyi, 2015; Szilagyi, 2007; Aminzadeh et al., 2016). However, the changes in b imply a potential nonlinear characteristic of the CR (Han, 2008; Lintner et al., 2015). The observed values of $E/E_{po}$ and $E_{pa}/E_{po}$ even exhibit a positive correlation under wet conditions at several flux sites, which challenges the linear CR (Han and Tian, 2018). "

Line 246: "Han and Tian (2018) further validated the sigmoid feature": Please state

the work more detailed because there are still controversies on it.

Response: We revised this sentence as "Han and Tian (2018) further validated the sigmoid feature according to the much larger regression slopes of $E/E_{Pen}$ upon $E_{rad}/E_{Pen}$ in the middle stage than those in the other two stages with smaller or larger values of $E_{rad}/E_{Pen}$ by using 22 eddy covariance towers from the FLUXNET (Baldocchi et al., 2001) dataset which includes representative biomes of grasslands, croplands, shrublands, evergreen needleleaf forests, deciduous broadleaf forests, and wetlands."

Line 270: What is the essential difference between B15 and H12? Is "B15 inherits all three types of evaporation dated from the original CR"? Please rearrange these sentences.

Response: The two generalized complementary approaches, H12 and B15, are essentially different, with completely different normalized variables (Table 3). B15 inherits the concept of the three types of evaporation dated from the original CR, and its boundary conditions and analytical form are derived for $x_B = E_{po}/E_{pa}$ and $y_B = E/E_{pa}$. By contrast, H12 goes much further from the original CR. The boundary conditions and the analytical form of H12 are derived for $x_H = E_{rad}/E_{Pen}$ and $y_H = E/E_{Pen}$. We added a new subsection "Comparisons between the two generalized complementary approaches" to discuss the essential difference between B15 and H12. Please refer to subsection "3.4 Comparisons between the two generalized complementary approaches" for details.

Line 304: The varying characteristics of the PT coefficient should be introduced here

Response: We introduced it in the revised manuscript, as "the Priestley-Taylor coefficient varies with several factors, such as the relative transport efficiency of turbulent, or the surface/air temperature (Assouline et al., 2016; Szilagyi, 2014)"

Line 359: Brutsaert's recent work by using c=0 and varying PT coefficient should be added. Check Brutsaert et al. (2019).

Response: We added it in the revised manuscript

The Conclusion part could be improved. I wonder are there any outlooks for future studies on CR could be summarized using a few sentences here?

Response: We revised the conclusion part and added three points about future studies: 1) "further validation and application of the two types of generalized complementary functions are required with multiple data over the world"; 2) "be

carefully examined for its physical base of the boundary conditions under completely wet environment", and 3) "be integrated with other approaches to include the information of the land surface properly". Please refer to the revised manuscript for details.

**Response to Referee #2**

**General Comments**

The authors present a very detailed review on the studies and developments of the complementary relationship of evaporation. Although the review is very detailed and scientifically well supported on the existing literature, I think it is too heavy due to the load of parameters introduced and unexplained, the long list of studies mentioned and a weak coherency when enumerating the studies. Can the authors make this easier for the reader to read through?

   Response: In the revised manuscript, we tried our best to make it easier for the reader to read through the following aspects: 1) Section "2 Symmetric complementary relationship" and "3 Efforts in maintaining a linear complementary relationship" of the original manuscript was be combined to one section "Linear complementary relationship" to make it more coherent. 2) Subsection "4.5 Improved understanding on the correlation between actual and potential evaporation" of the original manuscript was deleted by considering it is no tightly related to the whole review. 3) The parameters was checked and introduced more clearly in the revised manuscript. 4) We tried to improve the presentation and English.

Furthermore, I am completely missing the incites and perspectives from the authors. It would be very nice to see the opinion from the authors regarding the benefits of the framework. I suggest an extra section discussing 1) the best approach according to the authors criteria, 2) the future of the CR for E estimation, and 3) a comparison highlighting the advantages, disadvantages and opportunities of using the CR principle against other methods of Evaporation estimation that are not mentioned here. After this, the review should be ready for publication.

   Response: In the revised manuscript, we tried to give a clear point of view on the different approaches: 1) "The asymmetric CR is a significant improvement of the symmetric CR"; 2) "The generalized complementary principle with earlier linear CRs as special cases has a more rigorous physical base (Brutsaert, 2015; Han and Tian, 2018b), and its methodology based on nonlinear functions is robust and effective."

   We compared the two generalized complementary functions (the sigmoid H2017 and polynomial B2015) in a new subsection "3.4 Comparisons between the two generalized complementary approaches".

   We added a new subsection "4.3 Integrating with other approaches for further development" to discuss the current practice and future development of the CR for E estimation, and to compare it with other methods (the Penman approach, the Budyko approach and others) on the advantages, disadvantages and opportunities of using the

CR principle. At last, we proposed a suggestion of integrating these approaches for a new generation of evaporation estimation method.

**Specific Comments**

L. 24 State if it is a positive or negative feedback.

Response: We revised this sentence as "this principle originated from the negative feedback of areal evaporation on evaporation demand (Bouchet, 1963)".

L. 27 To understand, so the differences between Epa and Epo is just that Epa is small and local and Epo large-scale? Can you provide more explanation on what these two variables really mean since they are so important for this discussion? Specially for understanding Figure 1.

Response: The major differences between *Epa* and *Epo* are that they correspond to different atmosphere characteristics. *Epa* corresponds to the atmosphere in contact with current non-saturated evaporating surface as the overpassing air is not affected by the small saturated surface, whereas the atmosphere corresponding to *Epo* is in contact with the large-scale saturated surface. Thus, the surface water availability can be detected from the relative magnitude of *Epa* and *Epo* (as shown in Figure 1), and *E* can be estimated without the knowledges of the surface. We provided more explanation in the revised manuscript (Lines 30-34).

L. 32 What complex formulations?

Response: The formulations of *Epa* and *Epo* are introduced in Subection 2.1 and 2.4 of the revised manuscript. We will add a tip ("which will be reviewed in more detail in the following sections") in the revised manuscript.

L. 35-39 Please rephrase, it is difficult to understand.
L. 38 If there is a Penman calculation variable Epen, then how do you estimate Epa and Epo, that is different from Penman. Please specify.

Response: It should be noted that $E_{pa}$ and $E_{po}$ are theoretical concepts, and need to be formulated when applying for practical problems. The generalized complementary function comes in two ways. Brutsaert (2015) adopted a polynomial function to describe the relationship between $E$, $E_{pa}$ and $E_{po}$, and suggested to formulate $E_{pa}$ and $E_{po}$ by using Penman's potential evaporation ($E_{Pen}$) and Preistley-Taylor's minimal advection evaporation to formulate, respectively. By contrast, Han and Tian (2018) abandoned the theoretical concept of $E_{pa}$ and $E_{po}$, yet used a sigmoid function to describe the relationship among $E$, Penman's potential evaporation ($E_{Pen}$), and its radiation term ($E_{rad}$), which can be directly used for practical problems. We revised this paragraph as:

"Recent studies have adopted the "generalized" complementary principle, which employs nonlinear functions instead of the linear CR (Han et al., 2012; Brutsaert, 2015; Han and Tian, 2018). The generalized complementary function comes in two ways, with the first abandons the theoretical concept of $E_{pa}$ and $E_{po}$ yet uses a sigmoid function to describe the relationship among $E$, Penman's potential evaporation ($E_{Pen}$), and its radiation term ($E_{rad}$) (Han and Tian, 2018; Han et al., 2012). By contrast, the other adopts a polynomial function to describe the relationship between $E$, $E_{pa}$ and $E_{po}$. However, $E_{pa}$ and $E_{po}$ need to be formulated before applying the polynomial function to practical problems (Brutsaert, 2015)."

Table 1. Nice table!, but refer to the Appendix for the unexplained parameters.
Response: We added it below the table.

L. 69 why "basin-wide water balance" results? You said before that Epa is from a "small saturated surface".
Response: Epa was derived from a hypothetical theoretical concept of "small saturated surface", which means that the "small saturated surface" does not affect the atmosphere. Thus, it is determined by the atmosphere corresponding to current unsaturated surface. In application, Epa is calculated by using the meteorological variables corresponding to current unsaturated surface, and is used to calculate the basin-wide actual evaporation.

L. 70-75 But have these estimates been validated in some way?
Response: Yes. We pointed out it in the revised manuscript.

L. 75 When they found that it is overestimating or underestimating E, how did these studies obtain the real E then?
Response: In the revised manuscript, we added the potential causes of the bias "measurement error, imperfect formulations of $E_{pa}$ and/or $E_{po}$, external energy sources, or even the nonlinear nature of the complementary principle were considered as potential causes of this bias (Qualls and Gultekin, 1997; Hobbins et al., 2001; Han et al., 2008, 2012)." There are two approaches to obtain the real E: 1) maintaining a linear complementary relationship through properly formulating $E_{pa}$ and/or $E_{po;}$ 2) using a nonlinear generalized complementary function. Please refer to section "2.4 Efforts in", and "3 Generalized complementary principle via nonlinear functions" for the detailed methods.

L. 78 IMPORTANT. Since you are constantly introducing many parameters related to actual or potential evaporation, please include in the appendix a detailed explanation on the difference between each E parameter. For instance, to know how Epan differs

from Epa.

Response: Generally speaking, $E_{pa}$ and $E_{po}$ are theoretical concepts, whereas Epan, Epen, Ept and others are the specifications of them. They are introduced in the appendix to "Three types of evaporation in CR" and "Specific formulations for $E_{pa}$

or $E_{po}$" respectively in the revised manuscript.

L. 88 Do you mean that they change in opposite directions with increasing water availability?

Response: Yes. We changed "while" to "whereas" to make the sentence more clear.

L95 "the governing changes"

Response: We changed it in the revised manuscript.

L. 98 Why does Morton say that it is unrealistic and does not have proof, and argue against it, since you are performing a review on the subject.

Response: Morton derived the CR by two assumptions: "(1) the net radiation will not change with the surface, and (2) the heat and vapor eddy transfer

characteristics are identical for $E$ and $E_{pa}$. Relaxing the second assumption of Morton (1983), Szilagyi (2001) derived the CR by using the mass conservation equation for water vapor. However, LeDrew (1979) argued that Morton's two assumption do not necessarily hold, and pointed out that the symmetric CR is physically unrealistic by using a diagnostic model of the energy fluxes within a closed system."

L. 113 You mention an asymmetry, but before you were talking about symmetry?

Response: We revised it to "linear CR".

L. 136 So when is E_PT different from Epo. In other words, more clarity between these terms.

Response: Please refer Lines 174-180 of the revised manuscript:

"In theory, $E_{po}$ is the potential evaporation when the land surface is saturated,

and should be calculated with a proper formula by using meteorological variables corresponding to the "potential" saturated surface. The Priestley-Taylor equation has been widely accepted to represent evaporation from extensive saturated surfaces by using meteorological variables corresponding to these saturated surfaces (Brutsaert, 1982; Priestley and Taylor, 1972). This way it was suggested to represent $E_{po}$

(Brutsaert and Stricker, 1979). However, in the AA approach, $E_{PT}$ is calculated by

the Priestley-Taylor equation using the atmospheric variables that correspond to the

current unsaturated surface. But the atmosphere in contact with the land surface will change if the land surface was brought into saturated (Morton, 1983; Brutsaert, 2015).

Thus, $E_{PT}$ is in reality a variable dependent on the meteorological variables at the time of calculation and does not represent the "true" $E_{po}$."

L. 135-156 IMPORTANT I find these paragraphs hard to read and somehow "boring". As in a review, it would be very good if you can try to articulate all the studies in a more consistent way so that it does not become a list of studies and references each with a brief explanation. Also, many, many terms that have not been previously explained, only in an appendix. As it is, the review paper is now more focus to experts in the CR that common hydrologists.

Response: We rephrased this paragraph to make is more consistent and easy to read. In the first paragraph, the first problem of using EPT to denote Epo is pointed, and is explained as "predicting the air temperature corresponding to the extensive saturated surface is critical for properly formulating $E_{po}$". Then, three works aiming to settle this problem are introduced one by one: Morton (1983), Szilagyi and Jozsa (2008), and Aminzadeh et al. (2016).

Next, another problem is pointed as "Advection is another factor influencing $E_{po}$, which could not adequately considered by $E_{PT}$ with an assumption of a minimal advection effect (Morton, 1983, 1975; Parlange and Katul, 1992)." The works of Morton (1983) and Parlange and Katul (1992) are introduced.

Section 3. I don't see the rationale behind the selection of the subtitles 3.1 and 3.2. A brief explanation is needed. Why these subtitles, I assume 3.1 are the symmetric approaches and 3.2 the asymmetric ones? Think on the reader that is reading this review.

Response: We combined section 2 and 3 and change the order of former subsection 3.1 and 3.2. We believe it would be more rationale following this order:
2.1 Concept of symmetric complementary relationship;
2.2 Proofs of symmetric CR;
2.3 Asymmetric linear CR as an extension;
2.4 Efforts in maintaining a linear complementary relationship through.
The new 2.4 is for both the symmetric and asymmetric CR.

L.164 so b=1 means symmetry?
Response: Yes. We will add this in the revised manuscript.

L. 167. "The asymmetric CR is widely used?" Can you make a paragraph saying in your point of view which approach is better and why, symmetric or asymmetric?
Response: We deleted this sentence and add a paragraph as:

"The asymmetric CR is a significant improvement of the symmetric CR, and the opposite changes of $E/E_{po}$ and $E_{pa}/E_{po}$ against $E/E_{pa}$ were treated as an enhanced illustration of the CR (Hu et al., 2018; Zhang et al., 2017; Ma et al., 2015; Brutsaert et al., 2019; Szilagyi, 2007). The performances on evaporation estimation are improved by calibrating the asymmetry parameter $b$ (Kahler and Brutsaert, 2006; Han et al., 2008; Huntington et al., 2011; Ma et al., 2015). Efforts have also made to calculate $b$ by using the meteorological variables, which enhance the predict ability of the CR (Szilagyi, 2015; Szilagyi, 2007; Aminzadeh et al., 2016). However, the changes in $b$ imply a potential nonlinear characteristic of the CR (Han, 2008; Lintner et al., 2015). The observed values of $E/E_{po}$ and $E_{pa}/E_{po}$ even exhibit a positive correlation under wet conditions at several flux sites, which challenges the linear CR (Han and Tian, 2018). "

Can you make a similar Table 1 but for the non-linear relationships? I think that you mention many approaches that are not included in Table 2.

Response: We added a Table for the nonlinear generalized complementary functions as:

**Table 2. Different analytical formulas for generalized complementary functions H12**

| Type | Formula[*] | References |
|------|---------|-----------|
| Linear | $y = \alpha(1 + \dfrac{1}{b})x - \dfrac{1}{b}$ | Brutsaert and Stricker (1979) |
| | $y = (1 + \omega)x$ | Lhomme and Guilioni (2010, 2006) |
| Sigmoid | $y = \dfrac{1}{1 + c_1 e^{d(1-x)}}$ | Granger (1989), Han et al. (2011) |
| | $y = \dfrac{1}{1 + m(\dfrac{1}{x} - 1)^n}$ | Han et al. (2012) |
| | $y = \dfrac{1}{1 + m(\dfrac{x_{max} - x}{x - x_{min}})^n}$ | Han and Tian (2018) |
| Concave | $y = \dfrac{1}{1 + k(\dfrac{1}{x} - 1) + l}$ | Katerji and Perrier (1983), Han et al. (2014b) |
| | $y = (2 - c)\alpha^2 x^2 - (1 - 2c)\alpha^3 x^3 - c\alpha^4 x^4$ | Brutsaert (2015) |

[*] $x = E_{rad}/E_{Pen}$ and $y = E/E_{Pen}$. the other symbols are parameters.

**Response to Referee #3**

The authors provided an informative yet in-depth review of the research activities over the past half-century using the complementary principle of evaporation. I enjoyed the reading and feel it is overall a timely and nice contribution to the hydrology community. Nevertheless, I do challenge the authors to elevate the presentation quality, in a spirit to make it valuable for a variety of audiences, from those who are not very familiar with this type of research but would like to have some background information, to those who are really active in the field. In its current form, the writing style is still, more or less, in favor of the latter.

Response: Thanks for the positive comment to our manuscript.

We tried our best to make the review easier for the reader to read through. In the revised manuscript, we added a new subsection "Integrating with other approaches for further development" to compare the complementary approach of evaporation estimation with the approaches which are more popular in the research community, such as the Penman approach, the Budyko approach et., and propose a suggestion of integrating these approaches with the complementary principle for evaporation estimation. We think this new section would help the readers who are not very familiar with the complementary principle, but are interested in evaporation research.

Also, I feel it might be more straightforward to include the word "review" in the title and abstract. Otherwise, some readers may not realize this is a review article until the end of the introduction.

[revised manuscript text omitted]
 is widely used, and the parameter $b$ was considered a calibrated parameter at first (Kahler and Brutsaert, 2006; Han et al., 2008; Huntington et al., 2011; Ma et al., 2015a), but was calculated by using the meteorological variables (Szilagyi, 2015; Szilagyi, 2007; Aminzadeh et al., 2016).

The asymmetric CR can be illustrated in a dimensionless form (Figure 2) (Kahler and Brutsaert, 2006). Normalized by $E_{po}$, $E_{pa}$ and $E$ can be scaled as

$$\frac{E}{E_{po}} = \frac{(1+b)\,E/E_{pa}}{1+b\,E/E_{pa}} \text{ and } \frac{E_{pa}}{E_{po}} = \frac{1+b}{1+b\,E/E_{pa}}. \tag{4}$$

The scaled $E_{pa}$ and $E$ are both functions of the dimensionless variable $E/E_{pa}$, while $E/E_{pa}$ serves as the evaporative surface moisture index. Compared with the original form (Eq. (1) and Figure 1), the CR here is illustrated without the appearance of the water availability explicitly. The opposite changes of and against were treated as an standard illustration of the CR (Hu et al., 2018; Zhang et al., 2017; Ma et al., 2015a; Brutsaert et al., 2019; Szilagyi, 2007), but the wet conditions where the two curves of and approach were seldom focused. However, the observed values of and exhibit a positive correlation at several flux sites, which challenges the CR under wet conditions (Han and Tian, 2018a).

[Figure]

Figure 2.

[revised manuscript text omitted]

---

## Author Response (AR2)

**Response to the Referee**

I thank the authors for addressing my comments. This manuscript has become a nice review of the complementary principle of evaporation and I think it is ready for publication. But before, one last minor issue. Can the authors briefly state from the beginning why would hydrologists use the complimentary principle instead of other hydro climatic frameworks such as for instance that of Budyko (1974) and (1952) to estimate evaporation/evapotranspiration? This would increase the value of the article even more.

Response: Thanks the Referee for the positive comments. We would like to say that the advancement of the complementary principle doesn't aim at replacing other hydro-climatic frameworks, but integrating 
[revised manuscript text omitted]